# Nucleocytoplasmic transport of active HER2 causes fractional escape from the DCIS-like state

Lixin Wang[1,4], B. Bishal Paudel [1,4], R. Anthony McKnight [1,2] & Kevin A. Janes [1,3] ✉

Activation of HER2/ErbB2 coincides with escape from ductal carcinoma in situ (DCIS) premalignancy and disrupts 3D organization of cultured breast-epithelial spheroids. The 3D phenotype is infrequent, however, and mechanisms for its incomplete penetrance have been elusive. Using inducible HER2/ErbB2–EGFR/ErbB1 heterodimers, we match phenotype penetrance to the frequency of co-occurring transcriptomic changes and uncover a reconfiguration in the karyopherin network regulating ErbB nucleocytoplasmic transport. Induction of the exportin CSE1L inhibits nuclear accumulation of ErbBs, whereas nuclear ErbBs silence the importin KPNA1 by inducing miR-205. When these negative feedbacks are incorporated into a validated systems model of nucleocytoplasmic transport, steady-state localization of ErbB cargo becomes ultrasensitive to initial CSE1L abundance. Erbb2-driven carcinomas with Cse1l deficiency outgrow less irregularly from mammary ducts, and NLS-attenuating mutants or variants of HER2 favor escape in 3D culture. We conclude here that adaptive nucleocytoplasmic relocalization of HER2 creates a systems-level molecular switch at the premalignant-to-malignant transition.

Focal amplification or upregulation of *ERBB2* occurs at the premalignant ductal carcinoma in situ (DCIS) stage[1,2] before breast cancer onset. However, as an early oncogenic event, *ERBB2* amplification by itself elevates DCIS risk quite modestly[2]. Repeated attempts at molecular stratification of DCIS have been inconclusive[3,4], leaving open the question of whether there are cell-autonomous processes that restrain *ERBB2*-amplified premalignancies within the duct. Part of the challenge in identifying such modifiers lies in the regulatory complexity of ERBB2[5], which heterodimerizes with multiple ErbB family members (including EGFR) and signals from different compartments within the cell. Receptor activation starts at the plasma membrane[6], persists in some form after internalization, and can continue to the inner nuclear membrane upon binding karyopherins[7]. Further, ErbB signaling is normally tempered by waves of short- and long-term transcriptional feedback[8]. When ErbB-induced pathways become aberrant and

chronic, the immediate effect of these feedback mechanisms may change and give rise to trajectories that are heterogeneous rather than homeostatic.

Chronic ErbB signaling sporadically disrupts multicellular organization in 3D culture. Synthetic activation of ErbBs in breast epithelia causes hyperproliferative, polarity-disrupted multiacini resembling microscopic DCIS escapees (DEs)[9–11]. Curiously, global ErbB signaling causes the DE phenotype in only a fraction of clones in the culture[9]. A mechanism for the incomplete penetrance of DEs has evaded conventional molecular-biology[12] and transcriptomic[13] approaches. The phenotype is reportedly not heritable[9], but neither is it completely random, because DE frequency is dependent on the specific ErbB dimer[9,10]. Genes that synergize with other oncoproteins in 3D culture do not with dimerized ErbBs[14], and independent drivers of DEs usually antagonize the ErbB-induced phenotype[15,16]. Other genes

[1]Department of Biomedical Engineering, University of Virginia, Charlottesville, VA, USA. [2]Olympus Veran Technologies, St. Louis, MO, USA. [3]Department of Biochemistry & Molecular Genetics, University of Virginia, Charlottesville, VA, USA. [4]These authors contributed equally: Lixin Wang, B. Bishal Paudel. ✉e-mail: kjanes@virginia.edu

synergistically enhance ErbB-induced DEs[17,18], but it is not clear that they are variably induced or activated when ErbBs are dimerized. Authentic effectors of ErbB-associated DEs thus remain undefined.

Here, we quantitatively related DE frequency to upstream ErbB-induced gene-expression frequencies in single 3D outgrowths and identified surprising alterations in karyopherin-mediated nucleocytoplasmic transport. Karyopherins comprise a set of importins and exportins that use chemical gradients of GTP-bound Ran to shuttle large cargo in and out of the nucleus[19]. Incorporating transport of ErbB cargo and their associated feedbacks creates a toggle-switch architecture whereby divergent outcomes arise from the endogenous variability of an export karyopherin. DCIS lesions situated above or below this switch have different propensities for disorganized outgrowth in vivo, suggesting a limiting step for HER2+ premalignancies.

## Results

### Activated B2B1 cells elicit an incompletely penetrant DE phenotype

We built upon prior work by engineering MCF10A-5E breast epithelial cells[20] with separate chimeric receptors containing the intracellular domain of either human EGFR[10] or rat Erbb2[9] (Fig. 1a). The extracellular NGFR domain of each chimera prevents signaling from endogenous ErbB ligands and receptors; FKBP and FRB domains specifically heterodimerize the carboxy termini upon addition of the synthetic ligand AP21967 (AP). Cells with ectopic NGFR-EGFR-FRB and NGFR-Erbb2-FKBP were selected subclonally for expression at levels comparable to

endogenous EGFR and ERBB2 in breast cancer lines (Supplementary Fig. 1a). The leading clone (B2B1) modulated ERK1/2 and STAT1 pathways and induced canonical ErbB target genes when treated with AP in 3D culture (Supplementary Fig. 1b–e). B2B1 cells thus provided a means to investigate ErbB responses initiated by a single, biologically relevant heterodimer pair[5].

Phenotypically, AP treatment uniformly increased outgrowth of B2B1 structures within 24 h and sustained it for at least 9 days (Supplementary Fig. 1f). However, consistent with published results[9,11], we found that only ~35% of outgrowths exhibited a DE morphology of lost polarity and sphericity (Fig. 1b, c). Although DEs were obvious at 72 h and detectable at 48 h after AP addition, there was no clear indication before that despite persistent signaling from heterodimerized ErbBs for at least 24 h (Fig. 1d, Supplementary Fig. 1b–d and Supplementary Movie 1). Earlier studies of feedback did not look beyond 3–4 h of ErbB activation, leaving open the question of what adaptations were involved and how multi-cell fates could diverge so markedly.

### Guilt-by-association identification of candidates

New approaches are emerging to link phenotype variability with upstream heterogeneities in cell state, but such methods are not yet ready to tackle multicellular phenomena like DEs. We adopted a premise similar to recent work[21,22] and its predecessors[23] by postulating that heterogeneous transcriptional states preceded and primed the emergence of DEs (Fig. 1e). To identify state heterogeneities among different outgrowths, we combined robust 10-cell transcriptomics

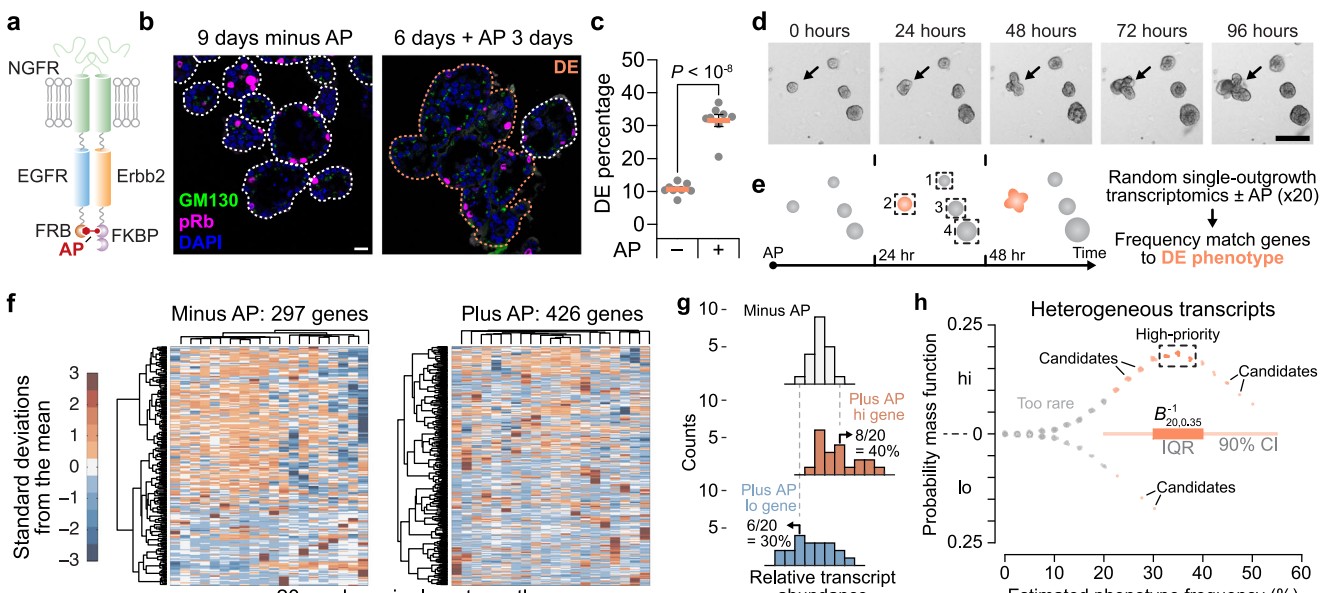

**Fig. 1 | Candidate regulators of a fractionally penetrant phenotype identified by stochastic frequency matching. a** Molecular design of NGFR-EGFR-FRB[10] and NGFR-Erbb2–FKBP[9] chimeras heterodimerized by AP21967 (AP). MCF10A-5E cells[20] were serially transduced–subcloned to generate B2B1 cells. **b** B2B1 cells were stained for Golgi polarity (GM130, green) and proliferation (phospho-Rb [pRb], magenta), counterstained for nuclei (DAPI, blue) and membrane-ECM (wheat germ agglutinin [WGA], white). One DCIS escapee (DE) in the plus-AP condition is outlined. Scale bar is 20 µm. **c** Fractional penetrance of the ErbB-induced DE phenotype in B2B1 cells. B2B1 cells were cultured in 3D for 9 days ±0.5 µM AP for the final 3 days. Data are shown as the arcsine transformed mean ± s.e. from n = 8 biological replicates where >200 outgrowths were scored per replicate. Differences were assessed by a two-sided t test after arcsine transformation. **d** DEs (arrow) are not apparent until after 48 h of ErbB heterodimerization. B2B1 cells were 3D cultured for 6 days, treated with 0.5 µM AP, and serially imaged by brightfield microscopy. Scale bar is 200 µm. **e** Approach for stochastic frequency matching. After 24 h of ErbB heterodimerization, 3D outgrowths (n = 20) were randomly sampled by laser-

capture microdissection and assessed by stochastic profiling[20,24]. Measured transcript fluctuations were compared to the DE frequency estimated in **c**. **f** Outgrowth-to-outgrowth transcript heterogeneities identified by stochastic profiling of B2B1 3D cultures ±0.5 µM AP for 24 h. Microarray probeset intensities were hierarchically clustered (Ward's linkage, row standardization). **g** Approach for quantifying transcript fluctuations. For each gene, the minus-AP condition (top) was used to specify 10th and 90th percentiles that score the number of outgrowths above (middle) and below (bottom) the minus-AP control for each gene. **h** Stochastic frequency matching gleans targets from outgrowth-to-outgrowth heterogeneities associated with ErbB heterodimerization. Probesets are jittered about the expectation for a binomial distribution (20 trials, 35% success probability) versus their estimated up-or-down expression frequency as in **g**. Probesets within the interquartile range (IQR) of the inverse binomial distribution ($B^{-1}_{20,0,35}$) were scored as high-priority, whereas those within the 90% confidence interval (CI) were considered candidates. Source data are provided as a Source Data file.

obtained by laser-capture microdissection with the principle of stochastic profiling for cell-state inference[20,24,25]. Here, each 10-cell sample was collected from a separate outgrowth (Fig. 1e, boxes) and restricted to cells of the periphery, recognizing that DE-relevant changes occur in this subpopulation[11]. We collected samples from 20 outgrowths with or without AP heterodimerization along with 16 pool-and-split 10-cell equivalents for each condition that control for technical variability[20,24,26]. Across (20 + 16) x 2 = 72 samples, we quantified 10,373 ± 1118 gene targets (mean ± s.d.) for analysis. Applying existing pipelines for distinguishing biological variability from technical noise[24], we predicted significant biological heterogeneity for 297 gene targets without AP and 426 gene targets with AP among B2B1 outgrowths (Fig. 1f and Supplementary Data 1 and Supplementary Data 2). We hypothesized that one or more drivers of the DE phenotype resided within this set of targets.

To winnow targets further, we leveraged the known 35% frequency of DEs anticipated to emerge 24 h after stochastic profiling was performed (Fig. 1c, e). We assumed that gene-expression states of DE effectors would also vary in ~35% of outgrowths at this time, considering binomial uncertainty from 20 outgrowths profiled per condition (Fig. 1g, h; see Methods). By requiring the stochastic frequencies of genes and phenotype to match, we created a stringent guilt-by-association filter for plausible upstream regulators of DEs. The stochastic frequency matching approach identified 15 high-priority targets within the interquartile range and another 86 candidates within the 90% confidence interval of the expected value of $0.35 \times 20 = 7$ (Supplementary Data 3). This pruning enabled biological classifications of the candidates and gene-by-gene perturbations of the high-priority targets.

## DE penetrance is perturbed by nucleocytoplasmic transport effectors

Nearly all genes that emerged from stochastic frequency matching populated 35% of the high-expression state (Fig. 1h, upper). For such targets, loss-of-function approaches were undesirable because reduced DE frequency might simply indicate perturbations that rendered outgrowths unfit. We thus adopted an inducible-overexpression approach to screen for induced targets that increase DE frequency. Genes were induced 1 day before AP addition to homogenize cells in the high-expression state when ErbBs were activated. By comparing 3D cultures ±AP and ±induction with doxycycline (DOX), we could test for two-way statistical interactions ($P_{int}$) between the induced target and the penetrance of DEs caused by ErbB heterodimerization.

Within the 15 high-priority targets lay XRCC6 and TTI1, raising the possibility that a heterogeneous DNA damage-like response was involved. Yet, induction of either gene had no detectable effect on AP-induced DEs compared to the minus-DOX condition or induced luciferase control (Supplementary Fig. 2a–c), prompting us to look more broadly. Gene ontology (GO) analysis of high-priority targets and candidates identified enriched GO Biological Processes related to localization and transport (Supplementary Fig. 2d). Within the enriched set, there were multiple genes connected to cytoskeletal trafficking: KIF20A, DYNLL2, MTUS1, and PFN2. However, inducing any of these genes did not promote more DEs but instead nearly abolished them (Supplementary Fig. 2e–h). The results excluded a pervasive role for cytoskeletal trafficking and suggested that loss of DEs could result from mechanistically unrelated stresses that act by simply blocking ErbB-induced hyperproliferation. Consistent with this notion, we arbitrarily induced BCAS2, a candidate tied to pre-mRNA splicing that resides within a breast-cancer amplicon[27], and found that it similarly eliminated DEs (Supplementary Fig. 2i).

We arrived at a different conclusion for a set of genes tied to nucleocytoplasmic transport (Fig. 2a–d). The exportin CSE1L scored as a high-priority target for two probesets and synergistically increased the frequency of DEs when induced (Fig. 2a). This result was further

complemented by loss-of-function studies in which inducible CSE1L knockdown significantly inhibited the AP stimulation of DEs (Fig. 2e and Supplementary Fig. 2j). Another high-priority target in this group was the nuclear pore-interacting protein NPIPB11 (originally annotated as LOC728888), which significantly elevated AP-independent DEs in B2B1 cells (Fig. 2b). Further work on NPIPB11 was precluded by difficulties in achieving reliable protein overexpression, as reported elsewhere[28]. The nuclear pore subunit NUP37 and the importin KPNB1 were both candidates by stochastic frequency matching and antagonized the ability of ErbB heterodimerization to induce DEs (Fig. 2c, d). The NUP37 effect was much more potent, and loss-of function studies argued against a generic stress-related defect upon perturbation, because inducible NUP37 knockdown significantly increased the frequency of basal DEs (Fig. 2f). Nevertheless, the directionality of NUP37 perturbations was opposite that predicted by stochastic frequency matching, suggesting a role as an attenuator of DEs instead of a driver like CSE1L.

Stochastic frequency matching accurately nominated upstream DE regulators that were unlikely to be uncovered by more conventional approaches. We measured transcript abundances in bulk 3D cultures at 24 h (when stochastic profiling was performed; Fig. 1e) or 72 h (when DEs were evident; Fig. 1d) and found surprisingly few differences with or without AP (Fig. 1g–j). Most genes of interest were not detectably altered, and the leading DE driver (CSE1L) was not increased >1.4-fold above minus-AP control even after 72 h of ErbB heterodimerization (Fig. 1g). Another karyopherin, the importin KPNA2, was more differentially abundant than CSE1L yet was not flagged as a cell-state heterogeneity and did not perturb DE frequency when ectopically induced (Supplementary Fig. 2k, l). We concluded that our approach was effective at revealing hidden drivers of incomplete penetrance and turned our attention to the general significance of CSE1L abundance for ErbB regulation.

## Cse1l promotes DE-like organization of Erbb2-amplified allografts

To go beyond B2B1 cells and ErbB activation with synthetic ligands, we sought a very different cancer-relevant setting to test the role of nucleocytoplasmic transport. We conjectured that AP heterodimerization of ErbBs resembles the early response of HER2+ cancers that have become resistant to ErbB-targeted therapeutics. This context was mimicked by examining release from cytostatic doses of the dual ErbB inhibitor, lapatinib, in TM15 clone 6 (TM15c6) cells derived from an Erbb2-amplified tumor of a $P_{MMTV}$-Cre;Erbb2$^{+/V660E}$ knockin mouse[29–31]. 3D cultures of TM15c6 cells were seeded in the presence of 2 µM lapatinib and grown for 6 days before lapatinib was washed out and outgrowth trajectories assessed. TM15c6 cultures did not exhibit DEs with all-or-none penetrance like those of nontransformed B2B1 cells, but lapatinib release caused significant decreases in circularity among the upper 35th percentile of outgrowths by size (Fig. 3a). Circularity is a scaled area-to-perimeter ratio that does not depend on geometric size, and we saw little evidence of any confounding area-circularity relationship in TM15c6 cultures for up to 5 days after lapatinib release (Supplementary Fig. 3a). These data established a foundation for genetic perturbations of Cse1l in TM15c6 cells.

We began with inducible ectopic expression of human CSE1L in TM15c6 cells (Supplementary Fig. 3b), using the same experimental paradigm as in B2B1 cells but substituting lapatinib release for AP addition. Relative to the pre-release circularity of each genotype, there was a reduction in circularity when CSE1L was co-induced with the washout of lapatinib (Fig. 3b). The CSE1L result was not heavily dependent on the upper percentile of outgrowths considered (Supplementary Fig. 3c). Reciprocally, inducible knockdown of Cse1l severely attenuated the loss of circularity from lapatinib release, and this phenotype was reverted upon co-induction of human CSE1L as an RNAi-resistant addback (Fig. 3c, d and Supplementary Fig. 3d).

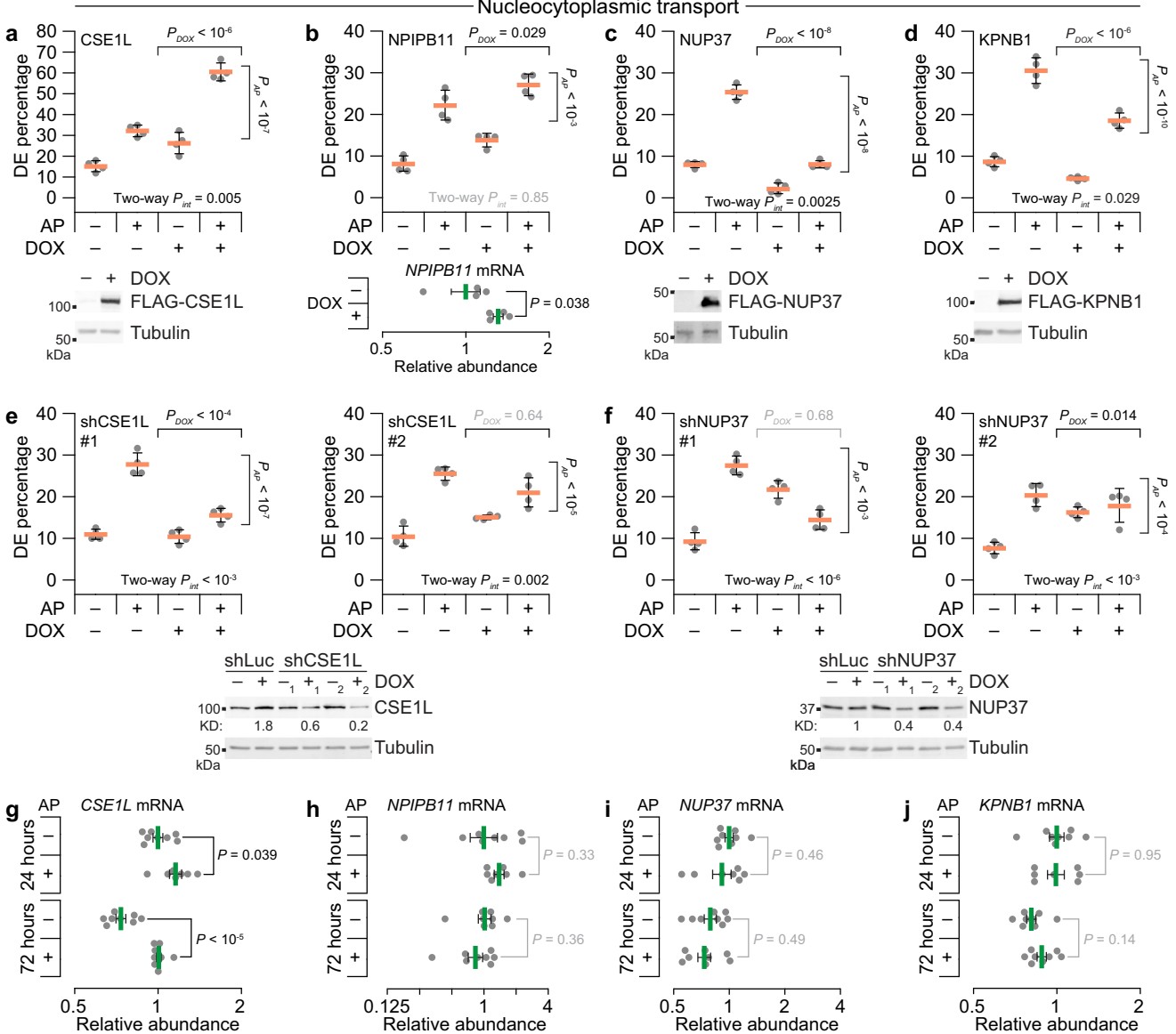

**Fig. 2 | Acute gain- and loss-of-function approaches support a role for fractionally induced effectors of nucleocytoplasmic transport imperceptible at the population level. a–d** Candidates related to nucleocytoplasmic transport affect DE penetrance induced by ErbB heterodimerization when ectopically expressed. B2B1 cells expressing inducible FLAG-tagged CSE1L (**a**), NPIPB11 (**b**), NUP37 (**c**), or KPNB1 (**d**) were used where indicated. **e, f** Loss of ErbB heterodimer-induced DEs upon inducible knockdown of nucleocytoplasmic regulators. B2B1 cells expressing inducible shCSE1L (**e**) or shNUP37 (**f**) were used where indicated. **g–j** Quantitative PCR for *CSE1L* (**g**), *NPIPB11* (**h**), *NUP37* (**i**), and *KPNB1* (**j**) in B2B1 cells cultured in 3D for 6 days followed by addition of 0.5 μM AP21967 (AP) for 24 h or 72 h where indicated. Data are shown as the geometric mean (normalized to the 24-h, minus-AP condition) ± log-transformed s.e. from $n = 7$ (plus-AP, 24 h) or 8 biological replicates

(all other groups). Differences in geometric means were assessed by two-sided *t* test after log transformation. For **a–f**, B2B1 cells stably expressing doxycycline (DOX)-inducible ectopic constructs were 3D cultured for 9–13 days with or without 0.5 μM AP added at Day 6 and/or 1 μg/ml DOX added at Day 5 where indicated. Data are shown as the arcsine transformed mean ± s.e. from $n = 4$ biological replicates where >100 outgrowths were scored per replicate. Differences by factor (DOX or AP) and two-factor interaction (int) were assessed by two-way ANOVA after arcsine transformation. Ectopic expression was confirmed by immunoblotting for FLAG with tubulin used as a loading control, except for *NPIPB11* where induction was confirmed by quantitative PCR. Inducible knockdown was quantified relative to the no DOX shRNA control targeting luciferase (shLuc). Source data are provided as a Source Data file.

Therefore, acute reactivation of ErbB signaling in TM15c6 mammary carcinoma cells elicits a Cse1l-dependent multi-cell phenotype that is reminiscent of DEs.

DCIS escape in vivo entails many additional barriers that are not reflected in 3D culture[32]. As a surrogate, we intraductally administered TM15c6 cells in a paired design comparing the macroscopic tumor characteristics of inducible shCse1l to contralateral control[33] after release from cytostatic doses of lapatinib (see Methods). Cse1l knockdown did not discernibly influence intraductal colonization or lesion stasis caused by 6 days of lapatinib treatment (Supplementary

Fig. 3e). At 2 weeks after lapatinib release, tumor bioluminescence was comparable between groups and both genotypes had invaded the stroma by this time. However, we posited that the tortuosity of the final tumors would reflect DE characteristics of TM15c6 cells at earlier stages. When glands were excised and tumors imaged ex vivo for higher resolution assessment of macroscale organization, we found that shCse1l tumors were consistently more circular than paired controls (Fig. 3e). Across the cohort, tumor circularity did not relate significantly to tumor mass (Supplementary Fig. 3f), excluding differences in net growth as a confounder. The TM15c6 study—with its

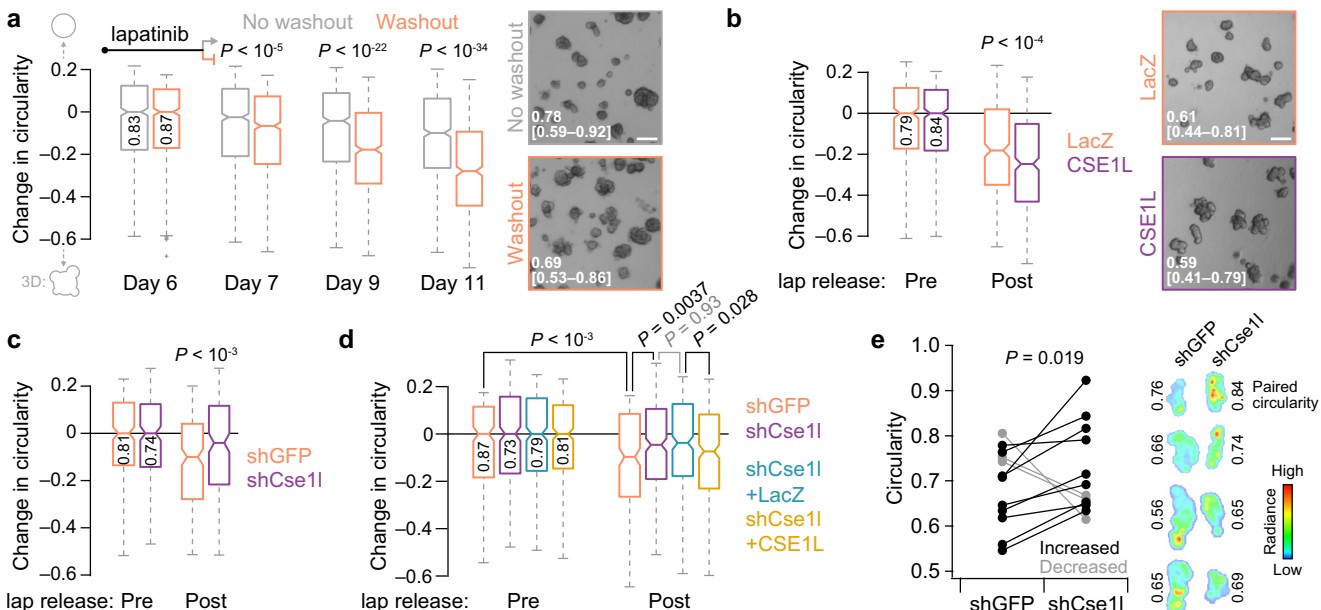

**Fig. 3 | *Erbb2*-amplified mammary carcinoma cells develop Cse1l-dependent DEs in vitro and in vivo after lapatinib release. a** Lapatinib release causes circularity loss in 3D-cultured mammary carcinoma cells with spontaneous *Erbb2* amplification. Representative brightfield images of 3D cultures ±lapatinib release for 3 days are shown on the right, with median circularity in the inset with interquartile range in brackets. **b** Ectopic induction of CSE1L exaggerates circularity loss following lapatinib (lap) release in 3D culture. Representative brightfield images of 3D cultures after 4 days of lapatinib release ±ectopic CSE1L are shown (right), with median circularity and interquartile range in brackets. **c**, Inducible knockdown of Cse1l inhibits the loss of circularity following lapatinib (lap) release in 3D culture. **d** Inducible addback of human CSE1L reverts circularity of shCse1l cells in 3D culture. **e** Paired change in circularity of lapatinib-released intraductal xenografts ±knockdown of Cse1l. Representative bioluminescence images of contralateral xenografts from four animals are shown on the right alongside the estimated overall circularity of each tumor. For **a**–**d**, TM15c6 cells stably expressing

doxycycline (DOX)-inducible CSE1L (or LacZ control) or shCse1l (or shGFP control) were 3D cultured for 5 days in 2 μM lapatinib, followed by 1 day in 2 μM lapatinib + 1 μg/ml DOX (**b**–**d**, pre) and then 1 μg/ml DOX for 1–5 days (**a**), 4 days (**b**, post), 3 days (**c**, post), or 1 day (**d**, post). Circularities were analyzed for the upper 35th percentile of outgrowths by size [*n* = (left to right) 751, 708, 922, 924, 859, 755 819, 636 outgrowths (**a**); 371, 476, 334, or 413 outgrowths (**b**); 242, 199, 208, or 173 outgrowths (**c**); 242, 292, 311, 316, 227, 301, 293, or 299 outgrowths (**d**) from four biological replicates; shGFP and shCse1l cells were used twice independently in **c** and **d**]. The 35th percentile was selected to match DE penetrance in B2B1 cells (Fig. 1c). Boxplots show the median reduction in circularity from the Day 6 time point (horizontal line, with Day 6 circularity reported vertically), estimated 95% confidence interval of the median (notches), interquartile range (box), and 1.5x the interquartile range from the box edge (whiskers). Differences between groups were assessed by two-sided rank sum test. For **a** and **b**, the scale bar is 200 μm. Source data are provided as a Source Data file.

change in species (human to mouse), ErbB activator (AP to lapatinib release), and cellular context (in vitro to in vivo)—strengthens the general conclusion that CSE1L/Cse1l drives DE-like phenotypes at the onset of persistent ErbB signaling.

## CSE1L-associated network states inhibit import of classical NLSs

The results with CSE1L, NUP37, NPIPB11, and KPNB1 in B2B1 cells (Fig. 2) justified examining other known regulators of nucleocytoplasmic transport. The pleiotropic exportin *XPO1* was detected as a biologically variable transcript by stochastic profiling in AP-treated cells but at a reduced expression frequency and with bulk changes that were muted compared to *CSE1L* (Supplementary Fig. 4a and Supplementary Data 3). Although not measurably heterogeneous, the GTPase *RAN* and its binding partners (*RANBP1* and *RANBP3*) were retained or elevated upon prolonged ErbB heterodimerization (Supplementary Fig. 4b–d). Conversely, factors regulating RanGTP hydrolysis (*RANGAP1*) and RanGDP exchange (*RCC1* and *NUTF2*) were not significantly altered, but independent bulk replicates of B2B1 cultures exhibited surprising dispersion (Supplementary Fig. 4e–g). We clustered the bulk transcript profiles by replicate and noted covariations among *KPNA2–KPNB1*, *CSE1L–XPO1*, and *RANBP1* that were nominally offsetting and thus difficult to interpret (Supplementary Fig. 4h). These observations suggested a need to examine nucleocytoplasmic transport at the systems level to determine the aggregate effect of the changes observed.

Drawing from earlier efforts[34,35], we built a multicompartment computational model encoding nuclear import and export modules with the Ran transport cycle (Fig. 4a). A complete description of the

model construction, scope, and parameter estimates is available in Supplementary Note 1 and Supplementary Data 4. For consistency, the model retains common-name designations from before[34,35]: CAS for CSE1L, Imp-α as a lumped parameter for KPNA1 through KPNA7, Imp-β for KPNB1, NTF2 for NUTF2, and CRM1 for XPO1. Changes in model parameters were not permitted unless a typographical error was discovered or the original parameter could not be definitively identified (Supplementary Note 1). To consider rate-limiting roles for NUP37 or NPIPB11, the model was elaborated with two classes of nuclear pores, one of which was selective to proteins harboring specific classes of nuclear localization sequences. However, this change proved inconsequential given the estimated number of nuclear pores per cell (Supplementary Note 1). The abundance of all protein species in the model was calibrated to B2B1 cells by quantitative immunoblotting against HeLa cells used in the prior work[34,35] to arrive at a draft model for validation (Supplementary Note 1 and Supplementary Data 4).

We tested the fidelity of the calibrated model by designing a series of inducible tandem cargo reporters harboring fluorescence proteins, epitope tags, and recognized sequences for import and export (Supplementary Fig. 4i). Reporters were individually induced at a few million copies per B2B1 cell and triply localized by immunostaining to mitigate the effect of cleavage products; then, images were aggregated and segmented for estimating nuclear-to-cytoplasmic (N/C) ratios at steady state in single cells (Supplementary Fig. 4i–k). All reporters contained an offsetting nuclear export sequence (NES) to distinguish the potency of each class of nuclear localization sequence (NLS)—a monopartite NLS from simian virus 40 T antigen (NLS_{SV40Tag}), a

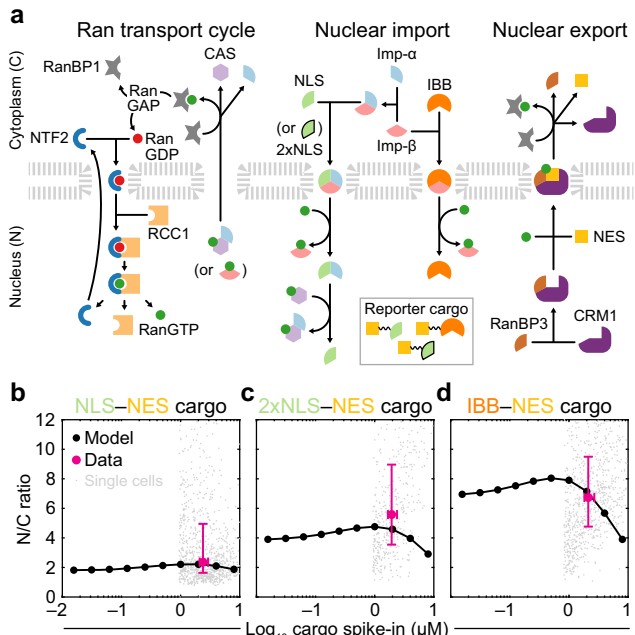

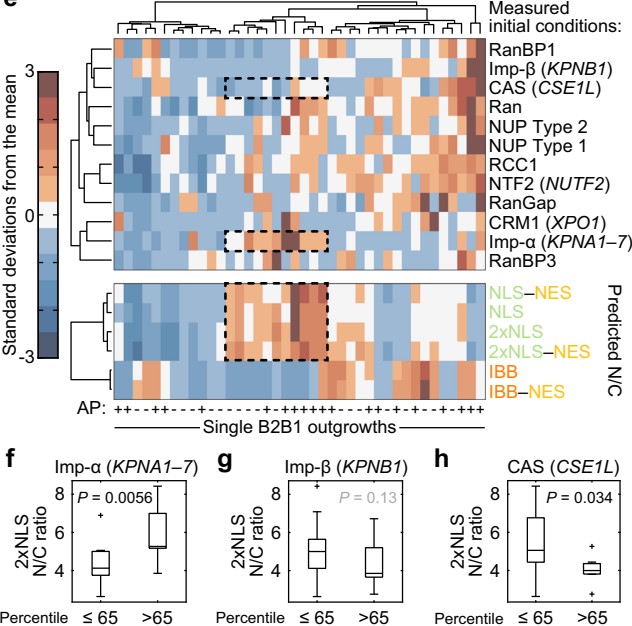

**Fig. 4 | ErbB heterodimerization triggers divergent nucleocytoplasmic transport. a** Overview of the nucleocytoplasmic transport model adapted from Riddick and Macara[34,35]. Arrows indicate the dominant direction of the reversible reaction. Reporter cargos were engineered to fuse a nuclear export sequence (NES) with sequences of simian virus 40 T antigen (NLS) or cap-binding protein 80 (2xNLS), which bind to Imp-α/β heterodimers with different affinity, or the Imp-β-binding domain of KPNA2 (IBB), which binds directly to Imp-β. See Supplementary Note 1. **b–d** Validated steady-state accumulation of induced NLS–NES (**b**), 2xNLS–NES (**c**), and IBB–NES (**d**) in the nucleus (N) relative to the cytoplasm (C). Steady-state N/C ratios were predicted for each reporter (black) and compared to population-level estimates of reporter abundance measured by quantitative immunoblotting[66] and N/C ratios measured by immunofluorescence (magenta). Population-level data (magenta) are shown as the mean ± s.e. from $n = 4$ (**b**, **c**) or 3 (**d**) biological replicates on the *x*-axis and the median ± interquartile range from $n = 1052$ (**b**), 445 (**c**), or 861 (**d**) cells (gray; median-scaled along the *x*-axis to the population mean) collected from four biological replicates. **e** Outgrowth-specific predictions of steady-state N/C ratios. Transcriptome-wide profiles from each B2B1 3D outgrowth ±AP21967 (AP, upper) were used to scale initial conditions (see Methods) and simulate steady-state accumulation of 1 μM NLS, 2xNLS, or IBB import sequences ±NES (lower). Model initial conditions and steady-state N/C ratios were hierarchically clustered separately by row and together by column (Ward's linkage, row standardization). Examples of high N/C states for NLS and 2xNLS are boxed alongside model species of interest. **f–h** Proportionally high expression of Imp-α and CAS, but not Imp-β, alters the steady-state accumulation of Imp-α/β-binding cargo. Predicted 2xNLS N/C ratio for $n = 20$ AP-treated B2B1 outgrowths split at the 65th percentile according to relative transcript abundance of *KPNA1–7* (**f**), *KPNB1* (**g**), or *CSE1L* (**h**). Boxplots show the median N/C ratio (horizontal line), interquartile range (box), 1.5x the interquartile range from the box edge (whiskers), and outliers (+). Differences between groups were assessed by one-sided rank sum test. Source data are provided as a Source Data file.

bipartite NLS from Cbp80 (2xNLS$_{Cbp80}$), or the Imp-β-binding domain of KPNA2 (IBB$_{KPNA2}$). NLS$_{SV40Tag}$ and 2xNLS$_{Cbp80}$ are recognized by Imp-α:Imp-β dimers with different affinity (K$_{D,SV40Tag}$ ~ 32 nM[36], K$_{D,Cbp80}$ ~ 2.5 nM[37]), whereas IBB$_{KPNA2}$ is imported after binding Imp-β alone[35] (K$_{D,IBB}$ ~ 6.5 nM[36]). We found that the tandem cargo reporters yielded different predictions of N/C ratio in the model, and these predictions compared favorably to measured values averaged across the population (Fig. 4b–d and Supplementary Fig. 4l). Because no recalibration was needed for any tandem cargo reporter, the experimental tests validated the model for provisional use.

Next, we predicted the steady-state N/C ratio as a readout of transport capacity for the different cargo classes when species abundances (initial conditions) of the model were perturbed combinatorially. Biologically realistic combinations were gleaned from the transcriptomic data collected among single B2B1 outgrowths during stochastic profiling (Fig. 4e). We found that presence of an NES had very little impact on the change of any reporter, consistent with its somewhat weaker potency[38] and the limiting abundance of the CRM1 cofactor RanBP3[39] in B2B1 cells (Supplementary Note 1). The predicted outgrowth-to-outgrowth patterns of NLS$_{SV40Tag}$ and 2xNLS$_{Cbp80}$ were also quite similar, owing to their same connectivity in the model (Fig. 4a, e). A cluster of perturbations with high predicted N/C ratio for these reporters was notable (Fig. 4e, dashed), and thus we stratified at the upper 35th percentile to ask whether any initial conditions were significantly associated. The analysis identified a significant positive association with Imp-α, but not Imp-β, and a

negative association with CAS/CSE1L (Fig. 4f–h). By contrast, IBB$_{KPNA2}$ engaged the network differently[35] and showed its own sensitivity to combinatorial perturbations, including a positive association for Imp-β and a negative association with CRM1/XPO1 (Supplementary Fig. 4m, n). The modeling results suggested a natural antagonism between karyopherins that depended on the type of cargo transported.

## CSE1L–NUP37 proximity interactomes contain ErbBs

It was centrally important to identify cargo that could be directly relevant to the DE phenotype. Because transport interactions are very transient, we developed a proximity-labeling approach focused on CSE1L and NUP37—two confirmed effectors that strongly perturb DE penetrance (Fig. 2a, c). Although *CSE1L* and *NUP37* transcripts are coexpressed broadly, there are no reports of binary or tertiary interactions involving these proteins. Thus, novel proximity interactions shared by CSE1L and NUP37 should strongly suggest conduits for the DE phenotype.

We scaled up from 3D culture to 15-cm plates by growing B2B1 cells as domes overlaid with dilute matrigel, a format that mimics 3D transcriptional changes for up to 10 days[40]. Lines expressing CSE1L or NUP37 fused to a promiscuous biotin ligase (BirA*) were moderately induced for 24 h (Supplementary Fig. 5a, b), followed by hetero-dimerization for 24 h and then biotin labeling for 24 h (Fig. 5a). Due to the overlay format, solubilized extracts contained many adsorptive extracellular matrix components that required a more-stringent two-

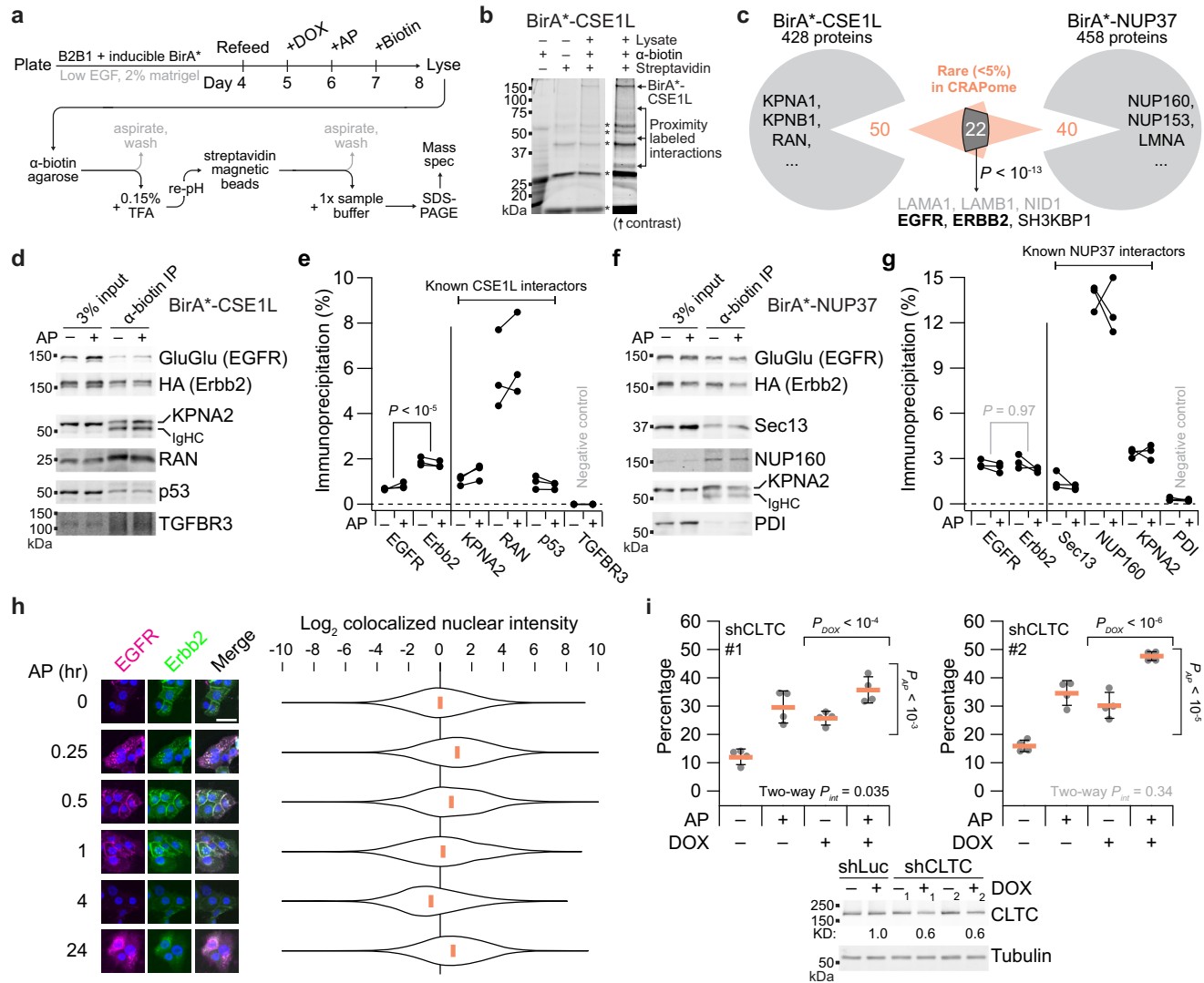

**Fig. 5 | CSE1L–NUP37 proximity labeling implicates a role for nucleocytoplasmic transport of EGFR–Erbb2. a** Overview experimental schema for proximity labeling with B2B1 cells. See Methods. **b** Two-step affinity purification of proximity-labeled extracts followed by SDS-PAGE. Non-specific bands from the terminal elution are indicated with asterisks. **c** Proximity-labeled proteins identified by mass spectrometry. Representative BioGRID interactors are listed along with notable shared targets based on their rarity in the CRAPome database[41]. Overlap significance was determined by one-sided hypergeometric test. Residual matrigel contaminants are listed in gray. **d**–**g** Replicated target validation for BirA*-CSE1L and BirA*-NUP37. Anti-biotin immunoprecipitates of proximity-labeled extracts from B2B1 cells expressing inducible FLAG-tagged BirA*-CSE1L (**d**) or BirA*-NUP37 (**f**) and treated ±0.5 μM AP for 24 h before biotinylation for 24 h were immunoblotted for the indicated targets and quantified from $n = 3$ paired biological replicates by densitometry (**e**, **g**). Differences in overall proximity labeling were assessed by two-way ANOVA with replication and AP treatment as a covariate. **h** Colocalized EGFR–Erbb2 puncta accumulate in the nucleus within 1 h and at 24 h after

heterodimerization. B2B1 cells were plated on coverslips, treated with or without 0.5 μM AP for the indicated times, and immunostained for EGFR–GluGlu (magenta) and Erbb2–HA (green) with DAPI counterstain for nuclei (blue). Scale bar is 20 μm. Violin plots show the median and distribution of nuclear-masked, colocalized GluGlu–HA pixel intensities from $n = 199$ cells (99,206 pixels [0 min]), 205 cells (100,125 pixels [0.25 h]), 192 cells (90,474 pixels [0.5 h]), 186 cells (88,002 pixels [1 h]), 229 cells (107,010 pixels [4 h]), and 198 cells (92,486 pixels [24 h]). **i** Knockdown of clathrin-dependent endocytosis elevates the basal penetrance of the DE phenotype. B2B1 cells expressing inducible shCLTC were 3D cultured for 10 days with 0.5 μM AP added at Day 6 and/or 1 μg/ml DOX added at Day 5 where indicated. Data are shown as the arcsine transformed mean ± s.e. from $n = 4$ biological replicates where >150 outgrowths were scored per replicate. Differences by factor (DOX or AP) and two-factor interaction (int) were assessed by two-way ANOVA after arcsine transformation. Inducible knockdown was quantified relative to the paired no DOX control with tubulin used as a loading control. Source data are provided as a Source Data file.

step purification of biotinylated target proteins (Fig. 5a). Sequential selection with anti-biotin and streptavidin yielded many proximity interactions detectable by mass spectrometry (MS) after SDS-PAGE purification of eluted products (Fig. 5b, c and Supplementary Data 5). The MS data contained multiple known interactors for CSE1L and NUP37 previously identified by co-immunoprecipitation, crosslinking, or yeast two-hybrid (Fig. 5c). We specifically confirmed literature-derived interactions by immunoblotting anti-biotin immunoprecipitates from independent samples: KPNA2, RAN, and p53 for CSE1L (Fig. 5d, e) and Sec13, NUP160, and KPNA2 for NUP37 (Fig. 5f, g).

Proteins sequestered from CSE1L or NUP37 were not detectably biotinylated, verifying the approach.

For uncovering novel proximity interactions in the MS data, we used a contaminant repository[41] to identify proteins that rarely appear in BirA* datasets (Fig. 5c and Supplementary Data 5). Among the dozens of proteins that passed the criterion, roughly half overlapped between the CSE1L and NUP37 datasets ($P < 10^{-13}$ by hypergeometric test). Strikingly, both studies identified EGFR, ERBB2/Erbb2, and the EGFR endocytic-trafficking adaptor SH3KBP1[42] as stringent proximity interactions (Fig. 5c and Supplementary Data 5). We independently

verified the MS result by immunoblots for the EGFR and Erbb2 chimeras, quantifying extents of proximity labeling in the range of known interactions (Fig. 5d–g). CSE1L was previously reported in MS data from EGFR immunoprecipitates[43]; we confirmed this interaction but found that Erbb2 was labeled much more efficiently than EGFR (Fig. 5d, e). The same was not true in B2B1 cells expressing BirA*-NUP37, where EGFR and Erbb2 were labeled equally well (Fig. 5f, g). This ruled out differences in chimera abundance as an explanation for the Erbb2-biased CSE1L result. CSE1L strongly increases DE penetrance (Fig. 2a), and EGFR homodimers lacking Erbb2 do not elicit a DE phenotype[9]. The data suggested that nucleocytoplasmic transport of Erbb2-containing dimers themselves might dictate penetrance of the phenotype they evoke.

### DE penetrance is inhibited by ErbB trafficking from the plasma membrane

Proximity labeling of EGFR–Erbb2 by BirA*-CSE1L or BirA*-NUP37 was not detectably altered by AP (Fig. 5d–g), excluding transport-protein binding as the regulatory step for ErbB-induced DEs. After ligand activation at the plasma membrane, ErbBs are internalized within a few minutes and can traffic to the nucleus in less than a half an hour[44]. We cultured B2B1 cells on coverslips and observed transient nuclear accumulation of colocalized EGFR–Erbb2 after 15 and 30 min of heterodimerization, which subsided at 1 h and disappeared at 4 h as the receptor was downregulated (Fig. 5h). Interestingly, after recovery in the presence of AP for 24 h, we quantified sustained EGFR–Erbb2 immunoreactivity in the nucleus that was nearly as intense as at early times after stimulation. This long-term nuclear localization coincided with the DE decision point (Fig. 1d, e), raising the possibility it could be relevant to phenotype penetrance. We tested the impact of ErbB internalization by mildly reducing clathrin heavy chain (CLTC) in B2B1 cells to impair clathrin-mediated endocytosis (Supplementary Fig. 5c). CLTC knockdown did not alter abundance of the ErbB chimeras, but CLTC-deficient cells exhibited significant increases in the frequency of DEs (Fig. 5i and Supplementary Fig. 5d), indicating that internalized EGFR–Erbb2 complexes are repressive.

### EGFR and Erbb2 chimeras are recognized as classical Imp-α:Imp-β cargo

Recruitment of ErbB family members to the nucleus occurs when a polybasic juxtamembrane segment within the ErbB sequence acts as a tripartite NLS for transport[44,45]. We returned to the systems model of nucleocytoplasmic transport (Fig. 4a) and asked how steady-state transport of different cargoes depended on the abundance of proteins in the network (Fig. 6a–e and Supplementary Fig. 6a–f). For the B2B1 model, NES-harboring cargoes were distinguished only when RanBP3 abundances were very high (Fig. 6a), allowing us to ignore consideration of ErbB–CRM1 interactions[46] in this setting. Further simplifying the analysis, we found that the sensitivities of monopartite (NLS_{SV40Tag}) and bipartite (2xNLS_{Cbp80}) NLS cargoes were identical except for an offset in N/C ratios from differences in affinity for Imp-α:Imp-β (Supplementary Data 4). By contrast, IBB_{KPNA2} cargo was more sensitive to CAS/CSE1L and Ran abundance, and transport qualitatively differed in response to the abundance of Imp-α and Imp-β (Fig. 6b–e). ErbB-family receptors bind Imp-β[46] but evidence for or against interactions with Imp-α was lacking. Systems-level analysis of ErbB nucleocytoplasmic transport thus required clarification of whether EGFR–Erbb2 heterodimers acted as classical NLS-like cargo or instead as non-classical IBB-like cargo[19].

To verify EGFR–Erbb2 interactions with Imp-β and test for interactions with Imp-α, we probed for epitope-tagged karyopherins in EGFR and EGFR–Erbb2 immunoprecipitates. B2B1 cells modestly expressing inducible KPNA2 (115 ± 10% of endogenous) or KPNB1 (80 ± 9% of endogenous) were probed for epitope tag after

immunoprecipitation of the EGFR chimera with or without prior AP treatment. We observed robust increases in co-immunoprecipitating Erbb2 chimera upon heterodimerization, as expected, and both KPNA2 and KPNB1 were consistently detectable in all immunoprecipitates (Fig. 6f, g). After scaling for target enrichment, there was no discernible difference in the level of KPNA2 vs. KPNB1 co-immunoprecipitation (Fig. 6g), suggesting the two karyopherins bind with roughly equal stoichiometry.

As a complementary approach, we evaluated potential interactions by proximity ligation assay (PLA). We confirmed assay specificity with positive and negative controls, as well as heterodimerizer-regulated proximity of EGFR and Erbb2 chimeras (Fig. 6h and Supplementary Fig. 6g, h). Above-background proximity ligation was observed for all ErbB-karyopherin pairs, with some signals approaching the saturated spot detection of the KPNA2:KPNB1 positive control (Fig. 6i, j). PLA spots often increased after extended heterodimerization from the increased stability of ErbB chimeras with AP (e.g., Fig. 5d). Together, we conclude that ErbBs behave like classical NLS cargo by binding an Imp-α:Imp-β heterodimer.

### Nuclear ErbBs engage a miR-205–KPNA1 negative-feedback circuit

Retrograde ErbB transport to the nucleus has been delineated mechanistically, but its consequences are multifaceted and contextual[7]. Nuclear ErbBs require their tyrosine kinase activity[47] and are reported to act as transcriptional coactivators[48] and chromatin-binding factors[44,47,48]. Therefore, we asked whether specific genomic loci were bound constitutively or inducibly to EGFR–Erbb2. By confocal imaging of 3D cryosections, we found EGFR–Erbb2 colocalized in the nucleus, and nuclear immunoreactivity increased significantly with the addition of AP for 24 h (Fig. 7a and Supplementary Fig. 7a). To preserve this 3D context as best as possible, input chromatin was prepared from B2B1 cells cultured in the overlay format for 1 week ± AP added on the sixth day. ErbB chimeras were immunoprecipitated by their respective C-terminal epitope tags and processed for chromatin immunoprecipitation sequencing (ChIP-seq) to identify enriched loci above naive IgG controls. For two different anti-epitope antibodies against the Erbb2 chimera, we immunoprecipitated comparatively little chromatin and were unable to detect any reliable enrichment peaks. This result is consistent with previous work involving ERBB2-amplified cells[48] and suggests that the C terminus is mostly obscured in the nucleus after fixation. ChIP-seq of B2B1 samples was successful when immunoprecipitating EGFR chimeras, and we recurrently identified enriched binding peaks for 26 loci without AP and 51 loci with AP, 17 of which were common to both conditions (Fig. 7b and Supplementary Data 6).

As reported[49,50], EGFR occupancy often occurred along the gene body of highly expressed genes (TXNIP, NEAT1) and transcription factors of the AP1 (JUNB) and KLF (KLF5) families. The AP-independent loci contained multiple genes previously linked to 3D cell-to-cell heterogeneity in MCF10A-5E cells: KRT5, NFKBIA, and SOD2[20,40,51]. These transcripts showed various time- and AP-dependent patterns of change, but none had been nominated as DE candidates by the earlier frequency-matching approach (Supplementary Fig. 7b–d and Supplementary Data 3). Instead, our attention turned to the MIR205HG locus encoding miR-205 (Fig. 7b), an epithelial microRNA frequently misregulated in cancer[52]. Although the earlier transcriptomic data could not be used to analyze miRNA abundance, miR-205 is known to be reduced by chronic ERBB2 signaling through changes in DNA methylation[53]. We confirmed miR-205 downregulation in B2B1 cells after 3 days of AP treatment but noted that the change was time dependent, reversing and becoming more variable when measured 2 days earlier (Supplementary Fig. 7e). Further, miR-205 abundance significantly increased when B2B1 cells were acutely heterodimerized in overlay cultures that improve uniformity of the response (Fig. 7c). As

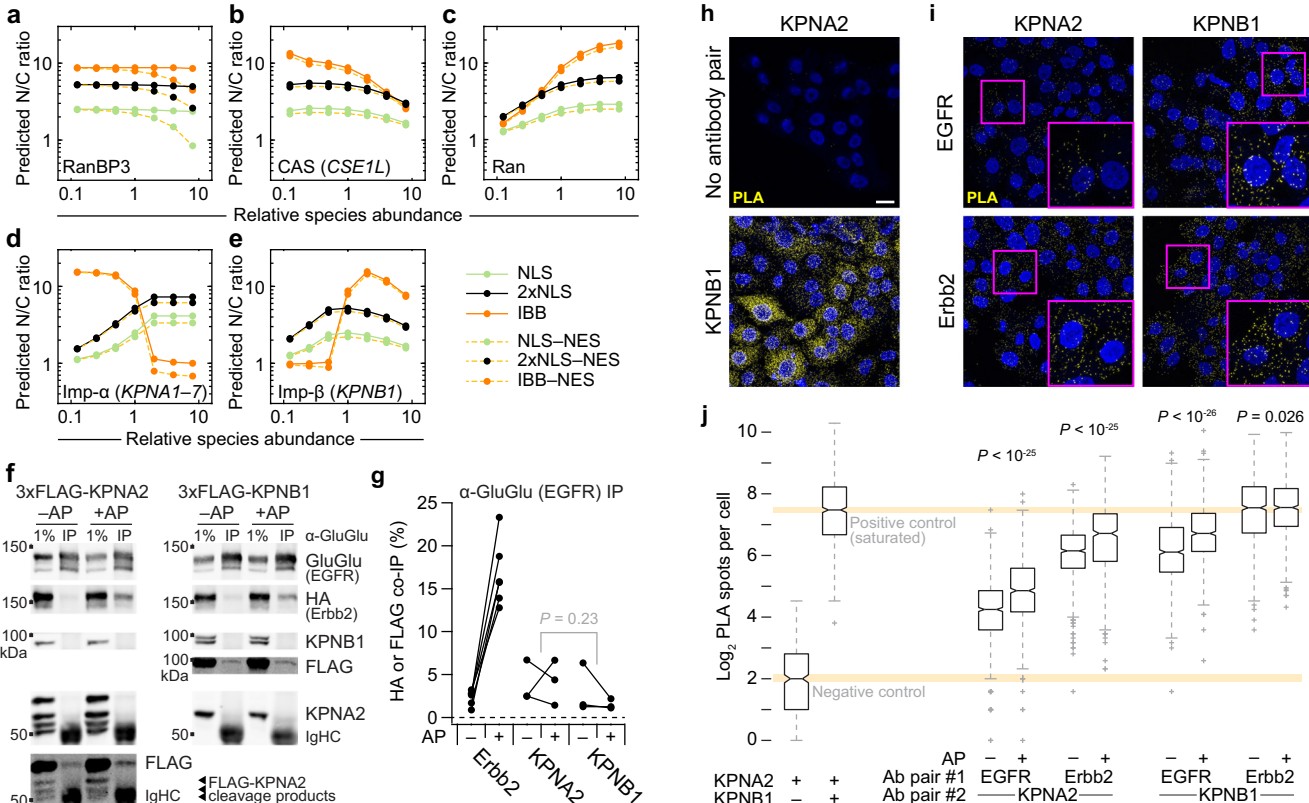

**Fig. 6 | EGFR and Erbb2 bind karyopherins of the classical nuclear-import family. a–e** Predicted sensitivity of the systems model to starting protein concentrations of RanBP3 (**a**), CAS (**b**), Ran (**c**), Imp-α (**d**), and Imp-β (**e**). Protein abundances are scaled relative to the concentrations used in the base model (Supplementary Data 4). Steady-state nuclear-to-cytoplasmic (N/C) ratios are shown for 1 μM of the representative cargo described in Fig. 4a. **f, g,** EGFR inducibly associates with Erbb2 and constitutively associates with KPNA2 and KPNB1. B2B1 cells expressing doxycycline (DOX)-inducible 3xFLAG-KPNA2 or 3xFLAG-KPNB1 were induced with 1 μg/ml DOX for 24 h and then treated with or without 0.5 μM AP for 1 h before lysis and GluGlu immunoprecipitation of the EGFR chimera. Samples were immunoblotted for the indicated targets (**f**) and quantified from $n = 3$ paired biological replicates by densitometry relative to the amount of immunoprecipitated EGFR chimera (**g**). Differences in overall KPNA2 and KPNB1 co-immunoprecipitation were assessed by two-way ANOVA with replication and AP

treatment as a covariate. **h, i** ErbB interactions with karyopherins confirmed by proximity ligation assay (PLA). B2B1 cells were plated and stained with the indicated antibody combinations. Negative and positive controls (**h**) were KPNA2 alone and KPNA2:KPNB1 respectively. Experimental antibody pairings (**i**) combined EGFR and Erbb2 chimeras with KPNA2 or KPNB1 as indicated. Scale bar is 20 μm. **j** Quantification of PLA spots in control samples and B2B1 cells treated with or without 0.5 μM AP for 24 h and stained with the indicated antibody combinations. Boxplots show the median PLA spots per cell (horizontal line), interquartile range (box), estimated 95% confidence interval of the median (notches), 1.5x the interquartile range from the box edge (whiskers), and outliers (+) from $n = 465$ cells (negative control), 856 cells (positive control), or (left to right) 960, 855, 849, 641, 985, 847, 934, or 781 cells from 16 confocal image stacks across two coverslips per condition. Differences between groups were assessed by two-sided KS test with Bonferroni correction. Source data are provided as a Source Data file.

with other loci, EGFR occupancy covered the *MIR205HG* promoter and length of the gene body (Fig. 7d). We used ChIP-qPCR to confirm specificity and AP-independence of binding (Fig. 7e); presumably, effects on the *MIR205HG* locus change when bound EGFR is co-associated with active Erbb2 (Fig. 7c). Analogously, EGFR–Erbb2 heterodimers strongly elicit a DE phenotype, whereas EGFR homodimers do not[9,10].

miR-205 posttranscriptionally regulates many genes[52], but one compelling target is the Imp-α family member *KPNA1*[54]. The 3' UTR of human *KPNA1* contains three predicted[55] binding sites for miR-205, the most proximal of which is broadly conserved, and we confirmed that miR-205 suppresses endogenous KPNA1 (Fig. 7f and Supplementary Fig. 7f)[54]. By immunofluorescence, KPNA1 protein expression was highly mosaic among single cells in B2B1 outgrowths (Fig. 7g). Importantly, KPNA1 protein abundance declined significantly upon addition of AP heterodimerizer without detectable changes in *KPNA1* mRNA, implicating a posttranscriptional mechanism (Fig. 7h and Supplementary Fig. 7g). When B2B1 cells were engineered with inducible ectopic KPNA1 lacking the native 3' UTR targeted by miR-205, we found that heterodimerizer-induced DEs were almost completely abolished (Fig. 7i). This result starkly contrasted earlier results with

KPNA2 that showed no effect upon ectopic expression (Supplementary Fig. 2i). Imp-α isoform switching has been linked to cell fate determination in other settings[56], and the miR-205–KPNA1 circuit induced by nuclear ErbBs added an isoform-specific negative feedback to the regulation of DEs.

## Nucleocytoplasmic ErbBs give rise to species-specific ultrasensitivity

The earlier systems model of nucleocytoplasmic transport predicted opposite effects of Imp-α and CAS/CSE1L abundance on the steady-state accumulation of classical cargo. Elevating Imp-α (e.g., KPNA1) increased N/C ratio of cargo and decreased DE penetrance (Figs. 4f and 7i), whereas high CAS/CSE1L decreased N/C ratio and increased DE penetrance (Figs. 2a and 4h). We thus modeled internalized ErbB heterodimers as classical cargo with an unknown affinity for Imp-α:Imp-β, equating high C/N [ = (N/C)$^{-1}$] ratio at steady state with the DE phenotype (Fig. 7j), presumably due to canonical signaling from cytoplasmic complexes. Imp-α and CAS/CSE1L abundances reciprocally affected C/N ratio across a wide range of binding strengths (Fig. 7j). However, model predictions were more sensitive to Imp-α than CSE1L except for very high abundances, and the changes in C/N

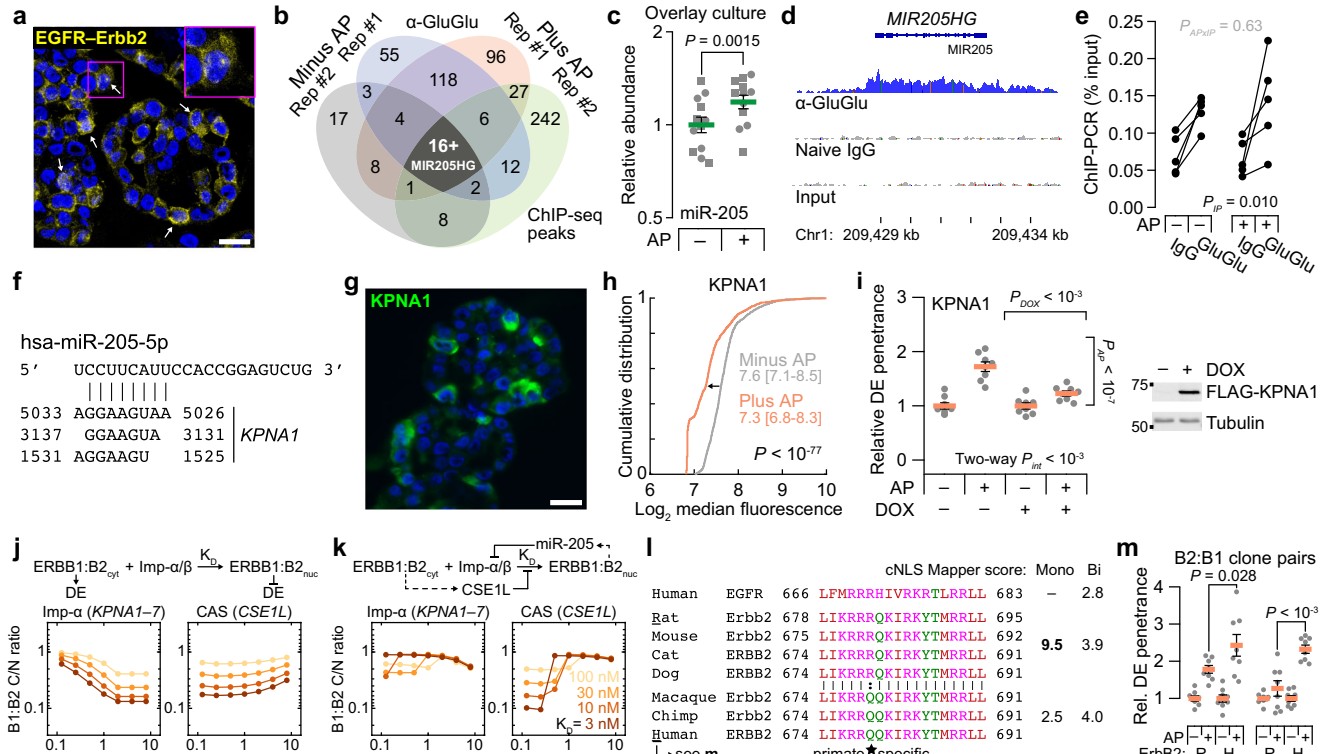

**Fig. 7 | Nuclear ErbBs induce miR-205–KPNA1 negative feedback causing species-specific, switchlike sensitivity to CSE1L. a** Colocalized nuclear EGFR:Erbb2 staining. GluGlu–HA images were quantile-normalized and geometrically averaged (yellow) with DAPI counterstain (blue). **b** Genomic peaks of EGFR-chimera ChIP-seq after overlay culture for 6 days ±0.5 μM AP for 24 h. **c** miR-205 abundance in B2B1 cells overlay-cultured for 6 days ±0.5 μM AP for 24 h. Data show the geometric mean ± log-transformed s.e. from *n* = 12 duplicated biological replicates (batches: circles, squares). Geometric-mean differences were assessed by two-way ANOVA (AP: fixed effect; duplicates: random effect) after log transformation. **d** Binding of EGFR chimera to the *MIR20SHG* locus. **e** ChIP-qPCR for *MIR20SHG* after overlay culture for 6 days ±0.5 μM AP for 24 h from *n* = 5 biological replicates. Three-way ANOVA assessed differences with IP antibody, AP, and biological replicates used as fixed effects. **f** TargetScan[55] predictions of miR-205 binding sites within *KPNA1*. **g** Immunofluorescence staining for KPNA1 (green) after 3D culture for 6 days +0.5 μM AP for 24 h plus DAPI counterstain (blue). **h** Median

KPNA1 immunoreactivity after 3D culture for 6 days ±0.5 μM AP for 24 h. Data show the cumulative distribution and median log₂ intensity [90% interval] from *n* = 894 (−AP) or 870 (+AP) cells from four biological replicates. Differences were assessed by two-sided KS test. **i** KPNA1 affects ErbB-induced outgrowth penetrance. Data are shown and analyzed as in Fig. 2a–d. **j**, **k** Model sensitivity to Imp-α−CSE1L abundance for ErbB heterodimers binding Imp-α:Imp-β with different dissociation constants (K_D). Simulations were performed with 1 μM cargo. **l** Tripartite NLS[45] alignment of ErbB orthologs. Groups are shown along with bioinformatic scores[57] as monopartite (Mono) or bipartite (Bi) NLSs. **m** B2:B1 clones expressing rat (R_1,2) or human (H_1,2) ErbB2 were 3D cultured for 10 days ±0.5 μM AP added at Day 5. DCIS escapee (DE) frequencies show the normalized mean ± s.e. from *n* = 8 biological replicates from >150 outgrowths. Differences in AP-induced DE penetrance were assessed by one-sided *t*-test. For **a** and **g**, the scale bar is 20 μm. Source data are provided as a Source Data file.

ratio were too gradual to explain how two DE states could coexist in the same population of cells.

We next considered the two antagonistic feedbacks elaborated before: (1) Active ErbBs internalize and traffic toward the nucleus, but they also induce *CSE1L*, which is predicted to inhibit ErbB nuclear transport by competing for available Imp-α (Figs. 2g, 4a, and 5h). (2) In parallel, nuclear ErbB heterodimers repress KPNA1 through miR-205, thereby reducing nuclear import, exacerbating the CSE1L competition with the Imp-α that remains, and increasing canonical cytoplasmic ErbB signaling (Fig. 7f–i and Supplementary Fig. 1b and 7f). After combining transcriptional, posttranscriptional, and cell-biological circuits, the overall ErbB feedback architecture (Fig. 7k, upper) is reminiscent of a synthetic-biology toggle switch capable of mixed single-cell behavior.

To determine whether cells were operating close to a mixed regime, we added proportional feedbacks from cytoplasmic ErbBs to CAS/CSE1L (positive) and from nuclear ErbBs to Imp-α (negative; Fig. 7k, upper). Repeating the simulations with the revised feedback architecture completely changed the dependence of C/N ratio on Imp-α and CAS/CSE1L. Sensitivity to overall Imp-α levels was severely blunted, consistent with results for KPNA2 (Supplementary Fig. 2i), and

the response to CSE1L became switchlike over a narrow range of expression (Fig. 7k, lower). We further noted that CSE1L abundance heterogenizes upon ErbB activation, with a range of expression that more than doubled after 24 h of AP addition (Supplementary Fig. 7h). The revised model indicates that very small initial differences in CSE1L are sufficient to yield divergent outcomes due to the nucleocytoplasmic transport properties of ErbB (Supplementary Fig. 7h, inset).

A key prediction of the ErbB transport model is that switch-like ultrasensitivity to CSE1L will decline when ErbB cargo affinity for Imp-α:Imp-β is reduced (Fig. 7k, right). In this scenario, the low-C/N, low-DE state is less dramatic, and prevalence of the DE phenotype should increase. We compared the tripartite NLSs from several extant species and found one R679Q variation, which was primate specific and predicted[57] to have weaker NLS potency (Fig. 7l). As tandem cargo reporters, the tripartite sequences of human EGFR and human ERBB2 were active but considerably weaker than that of rat Erbb2 (Supplementary Fig. 7i–k). To test the model prediction, we established independent EGFR-expressing clones wherein chimeric receptors of rat Erbb2 and human ERBB2 were comparably abundant (Supplementary Fig. 7l). For both rat-human pairs, DE penetrance was significantly increased for the human clone relative to the rat clone

(Fig. 7m). These results reinforce the model-driven assertion that DE and non-DE fates co-occur in ErbB-activated breast and mammary epithelia because of feedback regulation of active receptors and their transport to the nucleus.

## Discussion

This study unravels a set of time-evolving, competitive feedbacks that unfold hours to days after activation of ErbB2/HER2. CSE1L is a hallmark target gene of E2F, which becomes activated indirectly by cyclin D1 induction from immediate-early AP1-family transcription factors that are induced and stabilized by sustained ERK signaling from receptor tyrosine kinases. ErbB → CSE1L is slow because of the two rounds of signaling–transcription–translation required. Concurrently, active ErbBs are internalized and trafficked subcellularly, most toward endolysosomal degradation or recycling but some to alternative fates[44] including the nucleus. Various roles have been ascribed to nuclear ErbBs[7]. Our results are most concordant with work reporting ErbB-RNA polymerase II co-occupancy along the body of actively transcribed genes[50] such as *MIR205HG*. Here, ErbBs may directly impact the tyrosyl phosphorylation of RNA polymerase II and its subsequent transcriptional activity. miR-205 transcription, maturation, and translational repression is slow and multifaceted, yet human KPNA1 appears to be a particularly potent target impacting how internalized ErbBs are routed. An ErbB -| Imp-α relationship thus arises as a daisy chain of subcellular transport, signaling–transcription, and posttranscriptional regulation. Together, these feedbacks create a network architecture whose outcome is ultrasensitive to the initial abundance of CSE1L.

There are scattered reports of associations between HER2 signaling, nucleocytoplasmic transport, and CSE1L abundance in cancer. *ERBB2* and *CSE1L* amplifications are both frequent in medulloblastoma[58], and multiple HER2-positive breast cancer lines harbor separate gains in *CSE1L*[59]. As an exportin, CSE1L contains chemically reactive cysteines that are susceptible to covalent inhibition. CSE1L inhibitors might one day serve as adjuvants for HER2-amplified cancers, just as a first-in-class covalent inhibitor of the exportin XPO1 now enhances standard of care for multiple myeloma[60]. Besides CSE1L upregulation, other cancers may subvert the acute effects of nuclear ERBB2 directly by mutation. The tripartite NLS of ERBB2 resides in the juxtamembrane segment that is also responsible for positioning the tyrosine kinase for activation[6]. NLS-damaging mutations have been documented in Arg678, Arg677, Arg683, and Arg688 (Fig. 7l); importantly, an R678Q mutation observed in gastric cancer promotes DE-like growth of MCF10A cells[61]. From this standpoint, the primate-specific R679Q variant of ERBB2 may be an important consideration in reconciling debates surrounding the (patho)physiological importance of nuclear ErbBs.

We connected ErbB signaling to nucleocytoplasmic regulators by using a frequency-matching approach that statistically links transcript variability to phenotype variability. This guilt-by-association concept is effective when background fluctuations are mostly stochastic. Measuring 10-cell transcriptomes from single outgrowths enabled us to retain DE-relevant fluctuations while averaging out single-cell confounders such as cell-cycle status[26]. The method should be generally tractable for studying the molecular basis of incompletely penetrant phenotypes that are common and stereotyped, irrespective of the time scale on which they operate. There are several regulatory topologies conferring ultrasensitivity that remain to be discovered in biological systems.

## Methods
### Plasmids

**ErbB chimeras.** For EGFR, pBabe NGFR-FRB-GluGlu neo was prepared by triple ligation of pBabe neo digested with EcoRI–SalI, NGFR (1–274) PCR amplified and digested with MfeI–SpeI, and FRB-GluGlu PCR

amplified and digested with SpeI–SalI. Next, pBabe NGFR-EGFR-FRB-GluGlu neo was prepared by ligating pBabe NGFR-FRB-GluGlu neo digested with SpeI and EGFR (624–1165) PCR amplified and digested with XbaI–SpeI.

For Erbb2, pBabe NGFR-2xFKBP-HA hygro was prepared by triple ligation of pBabe hygro digested with EcoRI–SalI, NGFR (1–274) PCR amplified and digested with MfeI–XbaI, and 2xFKBP-HA PCR amplified and digested with XbaI–SalI. Next, pBabe NGFR-Erbb2-2xFKBP-HA hygro was prepared by ligating pBabe NGFR-2xFKBP-HA hygro digested with XbaI and Erbb2 (680–1259) PCR amplified and digested with SpeI.

For ERBB2, pBabe NGFR-ERBB2-2xFKBP-HA hygro was prepared by quadruple ligation of pBabe hygro digested with BamHI–SalI, NGFR (1–274) PCR amplified and digested with BamHI–XbaI, ERBB2 (676–1255) PCR amplified and digested with XbaI–MfeI, and 2xFKBP-HA PCR amplified and digested with MfeI–SalI.

**Inducible shRNA.** Doxycycline-regulated shRNAs from the RNAi Consortium were cloned into Tet-pLKO-puro (Addgene #21915) after changing the XhoI site in the shRNA loop to a PstI site as implemented previously[33]: shNUP37 v1 (TRCN0000152222), shNUP37 v2 (TRCN0000151312), shCSE1L v1 (TRCN0000061790), shCSE1L v2 (TRCN0000061792), shCse1l (TRCN0000217479), shCLTC v1 (TRCN0000380642), shCLTC v2 (TRCN0000379759). Control shRNAs targeted luciferase or GFP and were validated previously (Addgene #83092, #83085)[33].

**Inducible ectopic expression.** Doxycycline-regulated expression of BirA*, BCAS2, CSE1L, DYNLL2, KIF20A, KPNA1, KPNB1, MTUS1, NUP37, PFN2, TTI1, and XRCC6 were cloned by PCR into pEN_TT_miRc2 3xFLAG (Addgene #83274)[62]. For KPNA2, a linker was cloned into pEN_TT_miRc2 3xFLAG (Addgene #192305), and KPNA2 into pEN_TT_miRc2 3xFLAG linker. For NPIPB11-FLAG, a linker was cloned into pEN_TT_miRc2 (Addgene #25752), and NPIPB11-FLAG was cut with KpnI and SphI and cloned into pEN_TT_miRc2 linker. For proximity labeling, CSE1L and NUP37 were cloned by PCR into pEN_TT_miRc2 3xFLAG-BirA* (Addgene #192307). Full cloning details are available in Supplementary Data 7. All pEN_TT donor vectors were LR recombined with either pSLIK zeo (B2B1 cells; Addgene #25736) or pSLIK neo (TM15c6 cells; Addgene #25735) for lentiviral packaging. pSLIK 3xFLAG-Luciferase zeo (for B2B1 cells; Addgene #136533) or pSLIK 3xFLAG-LacZ neo (for TM15c6 cells; Addgene #83105) was used as a control[62]. All plasmids were verified by sequencing and deposited with Addgene (#192291–#192348).

**Tandem cargo reporters.** pEN_TT mCh-V5-NES-EGFP was prepared by triple ligation of pEN_TT_miRc2 digested with SpeI–MfeI, mCherry-ggtaagcctatccctaaccctctcctcggtctcgattctacg (V5) PCR amplified (without stop codon) and digested with SpeI–AgeI, and ttagccttgaaattag-caggtcttgatatc (NES)-EGFP-ggtacc (KpnI) PCR amplified (without stop codon) and digested with AgeI–MfeI. Tandem cargo reporters were then prepared by ligating pEN_TT mCh-V5-NES-EGFP digested with KpnI–MfeI and one of six nuclear localization sequences prepared with a stop codon and KpnI–MfeI overhangs: (i) NLS$_{SV40Tag}$—ccaaaaaagaaga-gaaaggta; (ii) 2xNLS$_{Cbp80}$—atgtcgcggaggcggcacagctacgagaacgatggtgga-caacctcacaaaaggaggaagacgtct; (iii) IBB$_{KPNA2}$—KPNA2 (1-70); (iv) NLS$_{EGFR}$—cgaaggcgccacatcgttcggaagcgcacgctgcggagg; (v) NLS$_{Erbb2}$—aaacgaaggagacagaagatccggaagtatacgatgcgtagg; (vi) NLS$_{ERBB2}$—aaac-gaaggcagcagaagatccggaagtatacgatgcgtagg. All reporters were LR recombined with pSLIK zeo (Addgene #25736) for lentiviral packaging.

### Cell lines and clones

For ErbB chimeras, MCF10A-5E cells[20] were transduced with pBabe NGFR-EGFR-FRB-GluGlu neo, selected with 300 µg/ml G418 (Sigma, A1720) until control plates had cleared, and cloned by limiting dilution

in 96-well plates to yield a founding B1 clone. B2B1 cells were isolated by transducing the B1 clone with pBabe NGFR-Erbb2-2xFKBP-HA hygro, selecting with 100 µg/ml hygromycin (Sigma, H0654) until control plates had cleared, and cloning by limiting dilution into 96-well plates. Paired human-rat clones were prepared by transducing the B1 clone with either pBabe NGFR-Erbb2-2xFKBP-HA hygro or pBabe NGFR-ERBB2-2xFKBP-HA hygro at titers adjusted to yield comparable HA expression at the population level. Then, polyclonal lines were surface labeled with NGFR antibody (NeoMarkers, MS-394-P0) and R-Phycoerythrin (PE)-conjugated goat-anti-mouse (Jackson ImmunoResearch, 115-116-146; 1:200 dilution), flow sorted into 96-well plates, and clones were screened for comparable receptor abundance by GluGlu and HA immunoblotting. TM15c6 cells (kindly provided by W.J. Muller)[29–31] were transduced with pLenti PGK V5-luciferase (w528-1) blast (Addgene #19166) and selected with 10 µg/ml blasticidin (Thermo Fisher, R21001) until control plates had cleared. HeLa cells used to calibrate the original nucleocytoplasmic transport model (kindly provided by I.G. Macara) were cultured in DMEM plus 5% fetal bovine serum, 5% newborn calf serum, and 1x penicillin-streptomycin.

### Viral transduction and selection
Lentiviral packaging was performed by calcium phosphate precipitation in 293 T cells with psPAX2 (Addgene #12260) and pMD2.G (Addgene #12259), with virus collected at 24 and 48 h, pooled, and passed through a 0.45 µm filter[40]. Viral transduction was performed with 8 µg/ml polybrene (Sigma, H9268), and cells were selected with 300 µg/ml G418 (Sigma, A1720) or 25 µg/ml zeocin (Invitrogen, 46-0509) until control plates had cleared[40].

### 3D culture
**Receptor activation.** For B2B1 cells and human-rat clones, cells were plated at 5000 cells per well on growth factor-reduced matrigel (Corning, 354230) in 8-well chamber slides (Falcon, 354108) and re-fed every 4 days according to published methods[63]. On Day 6, cultures were spiked with 0.5 µM AP21967 (Takara, 635057) or 0.1% ethanol vehicle, and these perturbations were retained in the culture medium during subsequent refeeds. For TM15c6 cells, cells were plated at 5000 cells per well on growth factor-reduced matrigel (Corning, 354230) in growth medium (DMEM medium (Gibco) plus 5% fetal bovine serum, 5 µg/ml insulin, 1 µg/ml hydrocortisone, 5 ng/ml EGF, 35 µg/ml bovine pituitary extract, and 50 µg/ml Gentamicin) supplemented with 2% matrigel + 2 µM lapatinib (MedChemExpress, HY-50898) and refed on Day 4. On Day 6, cultures were refed with growth medium + 2% matrigel and refed this way every 4 days thereafter.

**Phenotype scoring.** For B2B1 cells and human-rat clones, DEs were scored manually as 3D structures with at-least three lobes, excluding bi-lobed structures that could arise from two cells seeded next to one another. Total counts were collected by digital image acquisition and segmentation[64]. For TM15c6 cells, 3D structures were digitally acquired on an EVOS M7000 (ThermoFisher, AMF7000) with DiamondScope software (version 2.0.2094.0) and segmented to estimate circularity thresholded by overall size.

**Cryoembedding.** B2B1 cells were not compatible with prior immunostaining protocols[40,63] because outgrowths detached from the matrigel during the fixation step. Therefore, after washing briefly with PBS and fixing with 3.7% paraformaldehyde for 20 min at room temperature, the detached outgrowths from four wells were pooled, pelleted in a microcentrifuge tube, and washed three times in PBS for 5 min each. Fixed outgrowths were incubated in 15% (w/v in PBS) sucrose for 15 min and then 30% (w/v in PBS) sucrose for another 15 min before resuspension of the pellet in 200 µl yellow Neg-50 cryoembedding medium (Thermo Fisher, 6502Y). Samples were frozen in small cryomolds half-filled with frozen white Neg-50 cryoembedding medium (Thermo Fisher, 6502) and then surrounded with white Neg-50 immediately before freezing in a dry ice-isopentane bath. Cryosections were cut at 5 µm on a Cryostar NX50 cryostat (Epredia, 957130H) at −24 °C and air dried at room temperature before immunostaining.

### Overlay culture
B2B1 cells were seeded on tissue culture plastic (6-well to 15-cm plates) at 12,500 cells/cm² in assay medium + 2% matrigel + 5 ng/ml EGF (Peprotech, AF-100-15) and refed every 4 days. On Day 6, cultures were spiked with 0.5 µM AP21967 (Takara, 635057) or 0.1% ethanol vehicle for 24 h before cell lysis and biochemical analysis.

### Laser-capture microdissection and poly(A) cDNA amplification
Rapid histologic staining, laser-capture microdissection, and poly(A) cDNA amplification were performed according to published methods[20,24].

### Transcriptomic profiling
1 µg of poly(A)-amplified cDNA was labeled with Alexa Fluor 555 (Thermo Fisher, A32756) and hybridized to HumanHT-12 v4 BeadChip microarrays (Illumina, BD-901-1001) according to published methods[20,24]. Gene targets were considered present if the detection P-value for the probeset was < 0.1.

### Stochastic profiling analysis
BeadChip microarray fluorescence intensities were analyzed in MATLAB (R2022a) using StochProfMicroarrayFilt.m and StochProfAnalysis.m[24] with the following parameters: median detection threshold $P = 0.1$, maximum fold-change for reproducible detection = 5, false-discovery rate for sampling variation = 0.05, reference coefficient of variation = 0.2, and false-discovery rate for heterogeneity testing = 0.1.

### Stochastic frequency matching
For the 426 genes with significant heterogeneity in B2B1 cells plus AP, we compared single-outgrowth observations with the minus-AP condition to define a threshold for gene-expression state changes. The minus-AP data were used individually by gene to define empirical 10th and 90th percentiles for the background distribution. The threshold was motivated by the 10% background rate of DEs (Fig. 1c) attributable to the weak spontaneous dimerization of inactive ErbBs ($K_D$ ~ 37 µM)[65]. These percentiles were used as thresholds for the plus-AP observations to count single-outgrowth events above or below background. Taking the DE phenotype penetrance as a binomial probability ($p = 0.35$), we evaluated the probability mass function to determine how likely it was to see the observed number of events given the total number of observations: $B_{20,0.35}$. Reciprocally, we inverted the probability mass function ($B_{20,0.35}^{-1}$) to define an interval for the interquartile range of observations ([6–8] for $N = 20$) and the 90% confidence interval ([4–11] for $N = 20$). These ranges of events were used to define high-priority targets and candidates respectively.

### Immunoblotting
Total protein from 3D and 2D cultures was extracted with radio-immunoprecipitation buffer (50 mM Tris-HCl [pH 7.5], 150 mM NaCl, 1% Triton X-100, 0.5% sodium deoxycholate, 0.1% sodium dodecyl sulfate, 5 mM EDTA supplemented with 10 µg/ml aprotinin, 10 µg/ml leupeptin, 1 µg/ml pepstatin, 1 mM phenylmethylsulfonyl fluoride, 1 µg/ml microcystin-LR, and 200 mM sodium orthovanadate), separated on 8–12% polyacrylamide gels, transferred to polyvinylidene difluoride membranes, probed with primary antibodies (Supplementary Data 7) followed by IRDye-conjugated secondary antibodies, and scanned on a LI-COR Odyssey instrument with Odyssey software (version 3.0) as previously described[66].

## Bulk RNA extraction and purification

Total RNA from 3D and overlay cultures was extracted with RNA STAT-60 (Tel-Test) and purified according to the manufacturer's recommendation. Total RNA from traditional 2D cultures was extracted and purified with the RNEasy Plus Mini Kit (Qiagen, 74136) according to the manufacturer's recommendation.

## Bulk mRNA quantification

0.5–1 μg total RNA from 3D or overlay cultures was treated with Turbo DNAse (Thermo Fisher, AM2238), reverse transcribed with oligo(dT)$_{24}$ (IDT) and Superscript III (Thermo Fisher, 18080044), and specific transcripts were measured by quantitative PCR using 0.1 μl cDNA template and 1–10 pmol each of forward and reverse primers together with a homemade master mix used at a final concentration of 10 mM Tris-HCl (pH 8.3), 50 mM KCl, 4 mM MgCl$_2$, 200 mM each of dATP, dCTP, dGTP and dTTP, 150 μg/ml BSA, 5% glycerol, 0.25x SYBR green (Invitrogen, S7563), and 0.025 U/ml Taq polymerase (NEB, M0267) in a final reaction volume of 15 μl and published thermal cycling parameters[67]. Gene abundances were calibrated against purified amplicons and normalized to the geometric mean of four housekeeping genes: *GAPDH*, *GUSB*, *HINT1*, and *PRDX6*. Primer sequences and exact amounts used are listed in Supplementary Data 7.

## Intraductal xenografts

**Animals.** Female, virgin SCID/beige mice (Envigo, 186) were obtained at 7 weeks of age and housed together on a 10-h dark cycle (8 pm-6 am) and 14-h light cycle (6 am-8 pm) at 21.5 °C and 31.5% relative humidity. Only female animals were used because breast cancer is very rare in males. All animal work was done in compliance with ethical regulations under University of Virginia IACUC approval #3945, which permitted a maximal tumor size of 1.5 cm that was not exceeded by this study. For the experiments using doxycycline chow, mice were given a standard rodent diet containing 625 mg/kg doxycycline (Harlan, TD.05125, IF060, HF030) starting 1 week before surgery, which was maintained until the end of the study.

**Cell lines.** Luciferase-expressing TM15c6 cells stably expressing inducible shCse1l or shGFP were cultured in DMEM medium (Gibco) plus 5% fetal bovine serum, 5 μg/ml insulin (Sigma, I1882), 1 μg/ml hydrocortisone (Sigma, H0888), 5 ng/ml EGF (Peprotech, AF-100-15), 35 μg/ml bovine pituitary extract (Gibco, 13028-014), and 50 μg/ml Gentamicin (Gibco, 15750-060) at 37 °C with 1 μg/ml doxycycline added 2 days before inoculation.

**Inoculation and imaging.** Mice were anesthetized with isoflurane and the tip of each fourth and ninth mammary gland nipple was cut off with a fine scissor. Cells were suspended at a concentration of 20,000 cells/ml in growth medium plus 1 μg/ml doxycycline, and 2 μl of the cell suspension was loaded to a 30-gauge blunt needle (Hamilton, 80508) and then injected into the fourth mammary gland, with the contralateral gland receiving the paired control injection. Mouse surgical order and left-right glands assignments for experimental-control injections were randomized at the start of each experiment. Mice were imaged at 2, 5, 9, 12, 15, 18, and 22 days post-surgery on a Lago X (Spectral Instruments Imaging) with Aura Imaging Software (version 4.0.7) and after isoflurane anesthesia and intraperitoneal administration of 150 μg D-luciferin (Promega, E1605) per gram body weight to monitor bioluminescence. Five minutes after injection, images were acquired at 2-min intervals for 30 min with the following settings—1 s exposure; low (2) binning; F-stop = 1.2. From Days 3–8, mice were daily administered by oral gavage 30 mg/kg lapatinib (MedChemExpress, HY-50898) dissolved in 0.5% hydroxypropyl methylcellulose in water with 0.1% Tween 80. At the terminal endpoint, mammary tumors were excised and re-imaged for high-resolution bioluminescence to quantify tumor morphometry.

**Xenograft Image analysis.** Bioluminescence signals (radiance, photons/second/cm$^2$/steradian) were quantified using Aura Imaging Software (Spectral Instruments Imaging), and the maximum steady-state luminescences were recorded for subsequent analyses. Morphological outlines of excised tumors were quantified using ImageJ (version 1.53a).

## Nucleocytoplasmic transport model

A kinetic mass-action model of three compartments (cytoplasm, nucleus, and nuclear pore) and 207 rate equations was constructed in MATLAB (version R2022a) by recoding earlier efforts[34,35] and incorporating extensive changes to parameters and network topologies. Initial conditions for B2B1 cells were determined experimentally by quantitative immunoblotting against HeLa cells, which were previously used to obtain absolute estimates[34,35]. Complete description of the model is available in Supplementary Note 1.

## Proximity labeling mass spectrometry

**Labeling and sample preparation.** Four 15-cm plates of B2B1 cells stably expressing inducible BirA*-CSE1L or BirA*-NUP37 were prepared as overlay cultures, induced with 1 μg/ml doxycycline (Sigma, D9891) on Day 5, heterodimerized with 0.5 μM AP21967 (Takara, 635057) on Day 6, and labeled with 50 μM biotin (Sigma, B4639) on Day 7 for 24 h before lysis in RIPA buffer (50 mM Tris-HCl [pH 7.5], 150 mM NaCl, 1% Triton X-100, 0.5% sodium deoxycholate, 0.1% sodium dodecyl sulfate, 5 mM EDTA) plus protease and phosphatase inhibitors (10 μg/ml aprotinin, 10 μg/ml leupeptin, 1 μg/ml pepstatin, 1 mM PMSF, 200 μM Na$_3$VO$_4$). Clarified lysate (3 ml) was mixed with 8 ml IAP buffer (50 mM MOPS [pH 7.2], 10 mM sodium phosphate [pH 7.4], 50 mM NaCl) and immunoprecipitated with 500 μl anti-biotin agarose (ImmuneChem, ICP0615) overnight at 4 °C. Immunoprecipitates were washed four times with PBS plus protease and phosphatase inhibitors and eluted with 0.15% trifluoroacetic acid followed by re-equilibration to pH 7 with 1 M NaOH. The eluate was re-purified with 50 μl streptavidin magnetic beads (Thermo Fisher, PI88816) for 2 h at room temperature, washed three times with TBS + 0.1% Tween 20 plus protease and phosphatase inhibitors, and boiled in Laemmli sample buffer. Samples were electrophoresed on a 10% polyacrylamide gel and fragments excised to avoid nonspecific bands at 57, 41, 27, and 16 kDa (Fig. 5b) before mass spectrometry.

**Mass spectrometry.** Gel fragments (diced to ~1 mm$^3$ cubes) were transferred to a siliconized tube, washed in 200 μl 50% methanol, dehydrated in acetonitrile, rehydrated in 30 μl 10 mM dithiolthreitol in 0.1 M ammonium bicarbonate, and reduced at room temperature for 30 min. Samples were alkylated in 30 μl 50 mM iodoacetamide in 0.1 M ammonium bicarbonate at room temperature for 30 min, dehydrated in 100 μl acetonitrile, rehydrated in 100 μl 0.1 M ammonium bicarbonate, and then dehydrated again in 100 μl acetonitrile before drying completely by vacuum centrifugation. Gel fragments were rehydrated in 20 ng/μl trypsin (Promega, V5117) in 50 mM ammonium bicarbonate on ice for 30 min. Excess enzyme solution was removed, 20 μl 50 mM ammonium bicarbonate added, and the sample was digested overnight at 37 °C. Digested peptides were extracted in 100 μl 50% acetonitrile and 5% formic acid and evaporated to 15 μl for MS/MS analysis on a Thermo Electron Q Exactive HFX mass spectrometer system with an Easy Spray ion source connected to a Thermo 75 μm x 15 cm C18 Easy Spray column. 7 μl of the extract was injected and the peptides eluted from the column by an acetonitrile/0.1 M acetic acid gradient at a flow rate of 0.3 μl/min over 1 h. The nanospray ion source was operated at 1.9 kV. The digest was analyzed by acquiring a full scan mass spectrum to determine peptide molecular weights followed by product ion spectra (10 HCD) to determine amino acid sequence in sequential scans. The following parameters were used—S-lens RF 40, positive ions, MS [120 K resolution, 1 microscan, AGC 3e6, Max IT

60 ms], MS/MS [HCD, 30 K resolution, 1 microscan, AGC 1e5, Max IT 60 ms, min intensity 3.3e4, isolation 2.0, NCE 27, dynamic exclusion 20 s, no +1/unassigned], lockmass = 445.12006. The resulting RAW files were searched using Thermo Proteome Discoverer 2.4 using the embedded Sequest algorithm against the Uniprot Human Proteome (2/15/18, 93,273 entries) with the following parameters—10 ppm peptide, 0.02 Da fragments, full trypsin, carbamidomethyl Cys fixed, oxidized Met variable. Resulting search data files were processed in Scaffold (Proteome Software, ver 4.8.9) with the following parameters —Sequest score (+1 > 1.8, +2 > 2.0, +3 > 2.2, +4 > 3.0), peptide probability >60%, protein probability >90%, 1 peptide. The final data was compiled at a false-discovery rate of ~1%.

## Immunofluorescence

**Labeling and imaging.** Cells were plated at ~16,000 cells/cm$^2$ overnight on 22 × 22 mm No. 1.5 coverslips and the next day treated with 0.5 μM AP (Takara, 635057) for the indicated times where indicated. Cells were washed with 1 ml of PBS and fixed with freshly prepared 3.7% paraformaldehyde for 15 min at room temperature. Coverslips were washed with PBS three times for 5 min and blocked in a humid chamber for 1 h at room temperature in blocking solution (1x Western Blocking reagent [Roche, 11921673001] diluted in PBS and supplemented with 0.3% Triton X-100). Primary antibodies (Supplementary Data 7) diluted in blocking solution were added overnight at room temperature. The next day, coverslips were washed with PBS three times for 5 min, and Alexa Fluor 488, 555, or 647 secondary antibodies (Thermo Fisher, A-11001, A-11008, A-21422, A-21428, A-21235, A-21244) diluted 1:200 in blocking solution were added for 1 h at room temperature. Coverslips were then washed with PBS three times for 5 min, and counterstained with 0.5 μg/ml DAPI for 5 min. After two washes in PBS, coverslips were mounted with 10 μl 0.5% N-propyl gallate in 90% glycerol in 1x PBS, pH 8.0, and the edges were sealed with nail polish before imaging by widefield immunofluorescence on an Olympus BX51 microscope with Metamorph software (version 7.10.2.240) or a Leica STELLARIS 8 confocal microscope with LAS X STELLARIS control software (version 4.4.0.24861).

**Image analysis.** Immunofluorescence images were analyzed using custom codes written in MATLAB (version R2022a). Briefly, fluorescent images were background corrected using the Top-hat filtering function (imtophat), quantile-normalized, and combined by taking the geometric mean of signals to obtain a weighted measure of colocalized pixel intensities. DAPI was used to create the nuclear mask upon which colocalized signals within each nuclear mask were calculated. For cargo reporters, cells were additionally stained with 2 μg/ml wheat germ agglutinin (MP Biomedicals, 08790162) labeled with DyLight750 to create a whole-cell mask.

## Proximity ligation assay

**Labeling and imaging.** Cells were plated at 25,000 cells/cm$^2$ overnight on round 12 mm No. 1.5 coverslips in a 24-well plate overnight, treated with or without 0.5 μM AP for 24 h, and fixed–stained as with immunofluorescence up to the primary antibody step (Supplementary Data 7). Proximity ligation was performed with the Duolink In Situ Orange Starter Kit Mouse/Rabbit (Millipore Sigma, DUO92102-1KT) according to the manufacturer's recommendation. Samples were imaged in photon counting mode on a Leica STELLARIS 8 confocal microscope with a 40x objective (HC PL APO CS2 40x/1.30 OIL), 142 nm step size, 0.425 μs pixel dwell time per imaging track, 405 nm laser (1% power) excitation and 420–566 nm bandpass emission for DAPI, and 561 nm laser (0.3% power) excitation and 566–643 nm bandpass emission for Cy3. Eleven optical sections of 1 μm step size were collected from each image field and maximum-intensity projected for analysis.

**Image analysis.** Acquired images were processed using custom codes written in MATLAB (version R2022a) to obtain the number of spots per cell. Briefly, the raw images were filtered through a rotationally symmetric Laplacian of Gaussian filter (size = 15; standard deviation = 1), and a threshold of five photons per pixel was set to identify the number of spots in each image. Each spot above the threshold was assigned to the nearest nucleus to obtain the number of spots per cell. Nuclear staining with DAPI was used to obtain a nuclear mask, nuclear objects within 100–400 μm$^2$ were kept for further analyses.

## Chromatin immunoprecipitation (ChIP)

**Sample preparation.** Overlay cultures were prepared on 15-cm plates and harvested at Day 7 with or without 0.5 μM AP treatment for the preceding 24 h. Cells were crosslinked with 37% methanol-stabilized formaldehyde (Fisher Scientific, F79-500) at a final concentration of 1% for 10 min at room temperature. Crosslinking was quenched with 0.125 M Glycine for 5 min at room temperature. Cells were lysed with 300 μl lysis buffer (50 mM Tris-HCl [pH 8.0], 1% SDS, 5 mM EDTA, 10 μg/ml aprotinin, 10 μg/ml leupeptin, 1 μg/ml pepstatin, 1 mM PMSF). Chromatin was sonicated to an average size of 200–300 bp using a Bioruptor sonicator system (Diagenode) with 4 ×20 cycles of 30 s ON, and 30 s OFF at 4 °C. After sonication, a volume of 5% chromatin was set aside for input control, and the remaining chromatin was diluted tenfold in dilution buffer (20 mM Tris-HCl [pH 8.0], 1% Triton X-100, 2 mM EDTA, 150 mM NaCl, 10 μg/ml aprotinin, 10 μg/ml leupeptin, 1 μg/ ml pepstatin, 1 mM PMSF). Soluble chromatin was immuno-cleared with Protein A Plus Ultralink Resin (Thermo Scientific, 53135) for 2 h at 4 °C (50 μl for 1 ml chromatin). For chromatin immunoprecipitation, 5 μg antibodies or naive IgG control was added to the precleared chromatin and incubated overnight at 4 °C. Subsequently, 50 μl of Protein A Plus Ultralink Resin was added to each ChIP reaction and incubated for 4 h at 4 °C on a nutator. The pelleted beads were washed sequentially for 10 min on ice with 1 ml RIPA (once), 1 ml RIPA supplemented with 500 mM NaCl (three times), 1 ml LiCl buffer (10 mM Tris-HCl [pH 8.0], 1% NP-40, 1% sodium deoxycholate, 1 mM EDTA, 0.25 M LiCl) (twice), and 1 ml TE buffer (twice). The immunoprecipitated chromatin and withheld input chromatin were eluted and reverse crosslinked with elution buffer (10 mM Tris-HCl [pH 8.0], 0.5% SDS, 1 mM EDTA, 200 mM NaCl) to a final volume of 500 μl. Samples were then incubated with 100 μg/ml RNase (Qiagen, 1031301) for 30 min at 37 °C, followed by 200 μg/ml proteinase K (Invitrogen, 59895) for 90 min at 56 °C. Samples were purified by phenol-chloroform extraction and ethanol precipitation, and purified DNA was resuspended in EB buffer.

**Quantitative PCR.** 1 μl of purified ChIP DNA was diluted in 44 μl of H$_2$O. 4.5 μl of the diluted DNA was used for each PCR reaction. Primer sequences are listed in Supplementary Data 7. Signals from qPCR were quantified by ΔCt relative to input chromatin.

**Library preparation and sequencing.** Approximately 0.1–20 ng ChIP DNA was fragmented to a size range of 200 bp on a Covaris instrument for 2 min using 15 μl microTUBE-15 AFA Beads Screw-cap (Covaris 520145). Libraries were prepared with the NEBNext Ultra II DNA library preparation kit (New England Biolabs, E7645L), quantified by Qubit fluorimetry, and quality controlled on a Bioanalyzer. Samples were pooled for cBot amplification and sequenced as 75 bp paired-end reads on a NextSeq 500 (Illumina). Samples were demultiplexed with Bcl2fastq2 (version v2.20.0.422) before analysis.

**Analysis.** Read quality was assessed with FastQC (version 0.11.5), and reads were aligned to the human genome (GRCh38.86) using Bowtie2 (version 2.2.9) with default settings. Only uniquely mapped and non-duplicated reads were selected for further analysis. Peaks were called

using MACS2 (version 2.2.7.1) with default settings and comparing immunoprecipitated chromatin with input chromatin.

## *MIR205* quantification

5 ng total RNA from 3D or overlay cultures was used with the TaqMan MicroRNA Reverse Transcription Kit (Thermo Fisher, 4366596) and TaqMan MicroRNA Assays (Thermo Fisher, 4427975) for *MIR205* (Assay 000509) or *RNU6B* (Assay 001093) as a housekeeping snRNA. Abundances were quantified on a CFX96 Touch Real-Time PCR Detection System (Bio-Rad) with Bio-Rad CFX Manager 3.1 software (version 3.1.1517.0823) and using TaqMan Universal Master Mix II, no UNG (Thermo Fisher, 4440043) with the following thermal cycling protocol: 95 °C for 10 min, then 40 cycles of 95 °C for 15 s and 60 °C for 60 s. MIR205 quantification cycles were converted to ΔCt values upon subtraction of the paired *RNU6B* quantification cycle, which were then converted to ΔΔCt values upon subtraction of the average ΔCt value for the minus-AP control. Relative abundances are reported as 2-ΔΔCt values.

## Statistics and reproducibility

All statistical tests, their sidedness, and the number of replicates used for the test are included in the figure legends. For immunoblots that were confirmatory of protein expression, purification, or induction, single biological samples were analyzed with two (Fig. 5b), four (Supplementary Fig. 1a), seven (Supplementary Fig. 5a), or six (Supplementary Fig. 5b) technical replicates.

## Reporting summary

Further information on research design is available in the Nature Portfolio Reporting Summary linked to this article.

## Data availability

The raw and processed microarray data generated in this study have been deposited in NCBI's Gene Expression Omnibus and are accessible through GEO Series accession number GSE214455. The raw and processed ChIP-seq data generated in this study have been deposited in the Gene Expression Omnibus and are accessible through GEO Series accession number GSE216242. The proximity labeling proteomics data generated in this study have been deposited in ProteomeXchange under accession code PXD038327. The Genome Reference Consortium Human Build 38 version 86 (GRCh38.86) used in this study is available from Ensembl (ftp://ftp.ensembl.org/pub/release-86/gtf/homo_sapiens/Homo_sapiens.GRCh38.86.gtf.gz).

Source data for this work, including uncropped and unprocessed scans of immunoblots, are available on Figshare (https://doi.org/10.6084/m9.figshare.22179215.v1)[68].

## Code availability

The nucleocytoplasmic transport model has been deposited in the BioModels database (MODEL2210060001) and is available on GitHub (https://github.com/JanesLab/NucCytoShuttle)[69].

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

## Acknowledgements

We thank Gregory Riddick for early consultation on the nucleocytoplasmic transport model, Ian Macara for sharing the HeLa cell line, William Muller for sharing the TM15c6 cell line, Emily Farber and Suna Onengut-Gumuscu of the UVA Center for Public Health Genomics for assistance with BeadChip microarrays, Nicholas Sherman of the W.M. Keck Biomedical Mass Spectrometry Laboratory for assistance with mass spectrometry, Jun Lin for assistance with cloning, Zhao Lai of the Greehey Children's Cancer Research Institute UT Health San Antonio for assistance with the ChIP-seq experiments. We appreciate comments from Cameron Griffiths, Wisam Fares, Piotr Przanowski, and Róża Przanowska on the manuscript figures and Matthew Lazzara, Mohammad Fallahi-Sichani, and Bryce Paschal on the manuscript text. This work was supported by the NCI Cancer Systems Biology Consortium (U54-CA274499 and U01-CA215794 to K.A.J.), an NCI CURE Diversity Supplement (U01-CA215794-S1 to R.A.M.), the National Cancer Institute (R50-CA265089 to L.W. and P30-CA044579 to the UVA Comprehensive Cancer Center), and the David & Lucile Packard Foundation (#2009-34710 to K.A.J.). Sequencing data was generated in the Genome Sequencing Facility, which is supported by UTHSCSA, NIH-NCI P30-CA054174 (Cancer Center Support Grant at UTHSCSA), NIH Shared Instrument grant S10-OD030311 (S10 grant to NovaSeq 6000 System), and CPRIT Core Facility Award (RP160732). The W.M. Keck Biomedical Mass Spectrometry Laboratory is funded by a grant from the UVA School of Medicine.

## Author contributions

L.W. cloned and characterized B2B1 cells, performed the 10-cell profiling study, cloned genetic perturbations, and executed the proximity labeling study. B.B.P. cloned and characterized the rat-human pairs, performed and analyzed the ChIP-seq study, and analyzed immunofluorescence and PLA experiments. L.W. and B.B.P. together performed 3D experiments and in vivo experiments. R.A.M. built the nucleocytoplasmic transport model. B.B.P. and R.A.M. together performed and analyzed tandem cargo reporter experiments. K.A.J. analyzed data, edited the nucleocytoplasmic transport model, performed immunoblotting, qPCR, and PLA experiments, and drafted the manuscript. All authors edited the manuscript and secured funding to support this work.

## Competing interests

The authors declare no competing interests.
