## [Peer Review File · Nature Communications]

Nucleocytoplasmic transport of active HER2 causes fractional escape from the DCIS-like stateREVIEWER COMMENTS

Reviewer #1 (Remarks to the Author):

In this manuscript, the authors attempt to understand a long-standing dilemma in the context HER2 positive breast cancer. While high levels of HER2 overexpression are frequently observed in DCIS, there is a significant drop in frequency in metastatic lesions, raising a conundrum of why an oncogenic event that is a driver of tumor aggressiveness is lost during metastatic progression.

This study takes an elegant approach to identify a relationship between ErbB1/ErbB2 heterodimer activation and nucleocytoplasmic transport and its potential role in the progression of the DCIS state. There are multiple novel and important elements to the study, including the use of 10-cell transcriptomes in single outgrowth to reveal biology that otherwise would not be observed by bulk approaches; and the demonstrated relationship between ERB signaling and nucleocytoplasmic transport.

The major concern of the study is that the authors do not determine the observed relationship between ErbB1/ErbB2 and nucleocytoplasmic transport is a determinant of the stochastic nature of the DE state (the 35% of DE phenotype they observe). In particular, it is not clear if the regulation of nucleocytoplasmic transport is restricted to 35% of structures, and that is what regulates the partially penetrant phenotype or if there are alternative mechanisms that regulate the DE phenotype.

It is not clear why the authors postulate that the outer cells in the structure are likely the ones driving the DE phenotype when there have been multiple reports demonstrating significant intra-structure cell movement where cells inside change positions with outer cells rather fluently.

Minor concern: The authors could use significant shortening of the results sections by moving some of the technical details to the methods section to increase the readability of the manuscript. As presented, the manuscript will pose challenges to readability for the broad readership of Nat. Comm.

Reviewer #2 (Remarks to the Author):

In this study, the authors seek to find pre-existing variability in single cells leading to escape from ductal carcinoma in situ. They use a stochastic profiling approach to find genes whose variability in expression is of similar frequency (fraction of expressing cells) to the phenotype in question ("guilt by association"). Candidate factors identified by this approach were tested using overexpression panels, revealing the potential impact of nucleocytoplasmic transport in the escape process. They then delve more deeply into the molecular basis of the transport process, using proximity mass-spec to identify EGFR et al. The propose that nuclear localization of these signaling molecules triggers a microRNA feedback loop that then changes the DCIS escape phenotype.

Overall, I think this is a very impressive paper that moves the field forward. Single cell analysis in cancer has generally focused on areas like therapy resistance, so looking at processes involved in the oncogenic process is very exciting and forward looking. The frequency matching "guilt by association" method for gene identification is nice and could hopefully be applied more generally. The mechanistic follow up is truly remarkable, going a long way towards providing a molecular basis for the observed phenotype. It is rare and commendable to take genomic methods and push them all the way to "mechanistic biology". I should also say that I don't have expertise in a lot of the signaling work they did, so I will leave it to other colleagues to comment more extensively on that aspect of the manuscript.

I have a few comments of the take-it-or-leave it variety for this already very strong paper:

I would encourage the authors to explore ways to make the text more readable to a broad audience. I think there are central themes of broad interest, but they are to some extent buried underneath layers of jargon and technical description. It is of course the authors' prerogative to write the paper they like :), hence I provide this feedback just as a friendly suggestion.

The authors state: "We reasoned that it would be harder to increase DE frequency than to decrease it45 and thus adopted an inducible-overexpression approach to screen targets." I couldn't understand the reasoning for the overexpression approach based on this terse description, would be helpful to elaborate.

The authors state: "The results excluded a pervasive role for cytoskeletal trafficking and suggested that loss of DEs might generically indicate outgrowths that are unfit." I think what the authors are trying to say is that reduced fitness will just in general prevent outgrowths in ways that are not of mechanistic significance. Is that the case? Either way, would be good to elaborate on this point somewhat.

Fig. 7h: Can the authors report the effect size of the immunofluorescence experiment?

Regarding the proposed feedback loop mechanism for miR-205/KPNA1: it seems plausible, but I think there could be alternatives, given the complexity of the proposed mechanism. Can the authors describe some potential alternatives that can or cannot be eliminated with their data? To be clear, I do not expect the authors to do further experiments (in what is already a truly massive paper); rather, just outline some possibilities to evaluate against their existing data.

Arjun Raj

Reviewer #3 (Remarks to the Author):

This study investigates a karyopherin network regulating the nucleocytoplasmic transportation of ErbBs. The authors use a combination of techniques in this 'frequency-matching approach', but some further details are needed.

The authors report findings from the use of mass spectrometry to study proximity interactions, however absolutely no method details are provided on the MS.

- The authors need to detail the type of mass spectrometry used. There are dozens of variants of this technique and this needs to be detailed. Same with chromatography if this was an LC-MS method.

- The analytical parameters of the MS analysis need to be stated, including sample preparation, sample analysis, instrument settings.

- When discussing identification of proteins in the CSE1L and NUP37 datasets, the authors mention that dozens of proteins passed the criterion. What criteria were used for selecting proteins?

RESPONSE TO REVIEWERS

Reviewer #1:

In this manuscript, the authors attempt to understand a long-standing dilemma in the context of HER2 positive breast cancer. While high levels of HER2 overexpression are frequently observed in DCIS, there is a significant drop in frequency in metastatic lesions, raising a conundrum of why an oncogenic event that is a driver of tumor aggressiveness is lost during metastatic progression.

This study takes an elegant approach to identify a relationship between ErbB1/ErbB2 heterodimer activation and nucleocytoplasmic transport and its potential role in the progression of the DCIS state. There are multiple novel and important elements to the study, including the use of 10-cell transcriptomes in single outgrowth to reveal biology that otherwise would not be observed by bulk approaches; and the demonstrated relationship between ERB signaling and nucleocytoplasmic transport.

We thank the reviewer for their enthusiasm.

The major concern of the study is that the authors do not determine [whether] the observed relationship between ErbB1/ErbB2 and nucleocytoplasmic transport is a determinant of the stochastic nature of the DE state (the 35% of DE phenotype they observe). In particular, it is not clear if the regulation of nucleocytoplasmic transport is restricted to 35% of structures, and that is what regulates the partially penetrant phenotype or if there are alternative mechanisms that regulate the DE phenotype.

Although we cannot exclude all alternatives (see reply to Reviewer #2 below), our data are most consistent with a mechanism in which ErbB nucleocytoplasmic transport initiates similarly in all structures. Importantly, 3D-cultured cells exhibit a ~fourfold variation in CSE1L abundance before ErbBs are activated, and this variability exaggerates considerably with heterodimerizer (Supplementary Fig. 7h). The switch-like ultrasensitivity conferred by CSE1L and miR-205–KPNA1 feedbacks (Fig. 7k) suggested to us that cell-to-cell variation in CSE1L was sufficient to explain the divergent phenotype. In the revision, we tested this notion formally by using the feedback-encoded systems model (Fig. 7k) under a range of CSE1L initial conditions informed by single-cell estimates (Supplementary Fig. 7h). The modeling results predicted a 72-28 split in cytoplasmic/nuclear ratio of ErbBs, which compares favorably to the 65-35 split in DE phenotype. We have appended the simulations as an inset to Supplementary Fig. 7h in the resubmission (left).

It is not clear why the authors postulate that the outer cells in the structure are likely the ones driving the DE phenotype when there have been multiple reports demonstrating significant intra-structure cell movement where cells inside change positions with outer cells rather fluently.

Multicellular rotations and dynamic movements are indeed characteristic of early 3D culture, but they mostly disappear after the first four days¹. Previously, we used laser-capture microdissection to compare transcriptomes of inner and outer cells after six days of 3D culture and many differences related to matrix deprivation were reproducibly detected². The transcriptomic profiles in this manuscript were collected after seven days of 3D culture. Ras–ERK pathway activation reinitiates cellular movements within 3D spheroids³, and we wanted to capture such changes in cellular composition by restricting the \pm AP profiles to outer cells.

Minor concern: The authors could use significant shortening of the results sections by moving some of the technical details to the methods section to increase the readability of the manuscript. As presented, the manuscript will pose challenges to readability for the broad readership of Nat. Comm.

Based on this comment, we have considerably restructured the Abstract and Introduction based on feedback from three separate experts in ErbB signaling (Matthew Lazzara, UVA Chemical Engineering), cancer systems biology (Mohammad Fallahi-Sichani, UVA Biomedical Engineering), and nucleocytoplasmic transport (Bryce Paschal, UVA Biochemistry & Molecular Genetics). Further, we significantly shortened the main text description of stochastic frequency matching and the design of the TM15c6 in vivo experiment. The internal

reviewers also provided detailed comments on specific passages that were implemented to improve readability to a broad readership.

Reviewer #2:

In this study, the authors seek to find pre-existing variability in single cells leading to escape from ductal carcinoma in situ. They use a stochastic profiling approach to find genes whose variability in expression is of similar frequency (fraction of expressing cells) to the phenotype in question (“guilt by association”). Candidate factors identified by this approach were tested using overexpression panels, revealing the potential impact of nucleocytoplasmic transport in the escape process. They then delve more deeply into the molecular basis of the transport process, using proximity mass-spec to identify EGFR et al. They propose that nuclear localization of these signaling molecules triggers a microRNA feedback loop that then changes the DCIS escape phenotype.

Overall, I think this is a very impressive paper that moves the field forward. Single cell analysis in cancer has generally focused on areas like therapy resistance, so looking at processes involved in the oncogenic process is very exciting and forward looking. The frequency matching “guilt by association” method for gene identification is nice and could hopefully be applied more generally. The mechanistic follow up is truly remarkable, going a long way towards providing a molecular basis for the observed phenotype. It is rare and commendable to take genomic methods and push them all the way to “mechanistic biology”. I should also say that I don’t have expertise in a lot of the signaling work they did, so I will leave it to other colleagues to comment more extensively on that aspect of the manuscript.

We thank the reviewer for their enthusiasm.

I have a few comments of the take-it-or-leave-it variety for this already very strong paper:

I would encourage the authors to explore ways to make the text more readable to a broad audience. I think there are central themes of broad interest, but they are to some extent buried underneath layers of jargon and technical description. It is of course the authors’ prerogative to write the paper they like :), hence I provide this feedback just as a friendly suggestion.

In light of this comment, we recruited three separate experts in ErbB signaling (Matthew Lazzara, UVA Chemical Engineering), cancer systems biology (Mohammad Fallahi-Sichani, UVA Biomedical Engineering), and nucleocytoplasmic transport (Bryce Paschal, UVA Biochemistry & Molecular Genetics) to review the manuscript for readability. Their comments prompted extensive revisions to the main text, which are tracked in the revision.

The authors state: “We reasoned that it would be harder to increase DE frequency than to decrease it⁴⁵ and thus adopted an inducible-overexpression approach to screen targets.” I couldn’t understand the reasoning for the overexpression approach based on this terse description, would be helpful to elaborate.

Thank you, this passage was also marked as confusing by the three internal reviewers we recruited for the resubmission. We have revised the passage as follows:

Nearly all genes that emerged from stochastic frequency matching occupied high-expression states 35% of the time (Fig. 1h, upper). For such targets, loss-of-function approaches were undesirable because reduced DE frequency might simply indicate perturbations that rendered outgrowths unfit. We thus adopted an inducible-overexpression approach to screen for induced targets that increase DE frequency.

The review by Michael Kastan (Ref. ⁴⁵) that was originally cited describes this concept nicely. Because the text now stands alone, we have removed the citation from the resubmission to comply with journal reference limits.

The authors state: “The results excluded a pervasive role for cytoskeletal trafficking and suggested that loss of DEs might generically indicate outgrowths that are unfit.” I think what the authors are trying to say is that reduced fitness will just in general prevent outgrowths in ways that are not of mechanistic significance. Is that the case? Either way, would be good to elaborate on this point somewhat.

Yes, this was what we hoped to communicate. The revised text reads:

The results excluded a pervasive role for cytoskeletal trafficking and suggested that loss of DEs could result from mechanistically unrelated stresses that act by simply blocking ErbB-induced hyperproliferation.

Together with the revision above it, we hope that our message is clearer.

Fig. 7h: Can the authors report the effect size of the immunofluorescence experiment?

In the revised Fig. 7h, we add the median \log_2 fluorescence intensity and 90% interval. The relative \log_2 difference 0.35 or $1 - 2^{0.35} = \sim 27\%$, consistent with the magnitude expected from endogenous miRNA regulation.

Regarding the proposed feedback loop mechanism for miR-205/KPNA1: it seems plausible, but I think there could be alternatives, given the complexity of the proposed mechanism. Can the authors describe some potential alternatives that can or cannot be eliminated with their data? To be clear, I do not expect the authors to do further experiments (in what is already a truly massive paper); rather, just outline some possibilities to evaluate against their existing data.

We have mentioned two alternatives succinctly to the Discussion. First, we acknowledge that miR-205 has many targets in addition to *KPNA1*. Of note is a strong seed match to *CLTC*, which we showed increases DE frequency when knocked down with shRNA (Fig. 5i). An additional or alternate negative feedback involving miR-205 \rightarrow *CLTC* \rightarrow *ERBB1:B2_{cyt}* could change the toggle-switch properties demonstrated in Fig. 7k. Second, we clarify that the tripartite NLS in ErbBs also resides in a juxtamembrane that is critical for positioning the tyrosine kinase domain for allosteric activation⁴. Thus, NLS-damaging mutants could also impact the ability of ErbBs to activate or remain autoinhibited in the absence of ligand.

Reviewer #3:

This study investigates a karyopherin network regulating the nucleocytoplasmic transportation of ErbBs. The authors use a combination of techniques in this 'frequency-matching approach', but some further details are needed.

We have expanded the Methods section according to the comments of Reviewer #3 and provide specific responses below.

- *The authors report findings from the use of mass spectrometry to study proximity interactions, however absolutely no method details are provided on the MS.*
- *The authors need to detail the type of mass spectrometry used. There are dozens of variants of this technique and this needs to be detailed. Same with chromatography if this was an LC-MS method.*
- *The analytical parameters of the MS analysis need to be stated, including sample preparation, sample analysis, instrument settings.*

We apologize for the oversight. A few weeks after we submitted the original manuscript, we discovered that the proximity labeling section of the Methods had gone missing from the document we uploaded to *Nature Communications*. The revised manuscript includes the details that we mistakenly thought we had submitted in the first submission along with additional text edits provided from Dr. Nicholas Sherman from the UVA mass spectrometry core that further address the specific points raised by Reviewer #3.

- *When discussing identification of proteins in the CSE1L and NUP37 datasets, the authors mention that dozens of proteins passed the criterion. What criteria were used for selecting proteins?*

We schematized the criterion in Fig. 5c of the first submission: proteins needed to appear in both proximity labeling datasets (BirA*-CSE1L and BirA*-NUP37) and appear <5% in the proximity labeling CRAPome⁵. Identified proteins for both datasets and their corresponding CRAPome frequencies are listed in Supplementary Table 5.

REFERENCES

1. Wang H, Lacoche S, Huang L, Xue B, Muthuswamy SK. Rotational motion during three-dimensional morphogenesis of mammary epithelial acini relates to laminin matrix assembly. *Proc Natl Acad Sci U S A* **110**, 163-168 (2013).
2. Wang CC, Bajikar SS, Jamal L, Atkins KA, Janes KA. A time- and matrix-dependent TGFBR3-JUND-KRT5 regulatory circuit in single breast epithelial cells and basal-like premalignancies. *Nat Cell Biol* **16**, 345-356 (2014).
3. Pearson GW, Hunter T. Real-time imaging reveals that noninvasive mammary epithelial acini can contain motile cells. *J Cell Biol* **179**, 1555-1567 (2007).
4. Jura N, *et al.* Mechanism for activation of the EGF receptor catalytic domain by the juxtamembrane segment. *Cell* **137**, 1293-1307 (2009).
5. Mellacheruvu D, *et al.* The CRAPome: a contaminant repository for affinity purification-mass spectrometry data. *Nat Methods* **10**, 730-736 (2013).

REVIEWERS' COMMENTS

Reviewer #1 (Remarks to the Author):

The authors have addressed my concerns satisfactorily.

Reviewer #2 (Remarks to the Author):

The authors have done a nice job revising the manuscript. My concerns are addressed. Arjun Raj

Reviewer #3 (Remarks to the Author):

The authors have responded to my concerns accordingly. No further comments.

REVIEWERS' COMMENTS

Reviewer #1 (Remarks to the Author):

The authors have addressed my concerns satisfactorily.

Reviewer #2 (Remarks to the Author):

The authors have done a nice job revising the manuscript. My concerns are addressed. Arjun Raj

Reviewer #3 (Remarks to the Author):

The authors have responded to my concerns accordingly. No further comments.

We are happy to hear that all reviewer concerns have been addressed.